# In situ evidence of thermally induced rock breakdown widespread on Bennu's surface

J. L. Molaro [1✉], K. J. Walsh [2], E. R. Jawin[3], R.-L. Ballouz [4], C. A. Bennett [4], D. N. DellaGiustina [4], D. R. Golish [4], C. Drouet d'Aubigny[4], B. Rizk [4], S. R. Schwartz [4], R. D. Hanna [5], S. J. Martel [6], M. Pajola [7], H. Campins [8], A. J. Ryan [4], W. F. Bottke[2] & D. S. Lauretta[4]

Rock breakdown due to diurnal thermal cycling has been hypothesized to drive boulder degradation and regolith production on airless bodies. Numerous studies have invoked its importance in driving landscape evolution, yet morphological features produced by thermal fracture processes have never been definitively observed on an airless body, or any surface where other weathering mechanisms may be ruled out. The Origins, Spectral Interpretation, Resource Identification, and Security–Regolith Explorer (OSIRIS-REx) mission provides an opportunity to search for evidence of thermal breakdown and assess its significance on asteroid surfaces. Here we show boulder morphologies observed on Bennu that are consistent with terrestrial observations and models of fatigue-driven exfoliation and demonstrate how crack propagation via thermal stress can lead to their development. The rate and expression of this process will vary with asteroid composition and location, influencing how different bodies evolve and their apparent relative surface ages from space weathering and cratering records.

---

[1] Planetary Science Institute, 1700 E Ft Lowell Rd., STE 106, Tucson, AZ 85719, USA. [2] Southwest Research Institute, 1050 Walnut St #300, Boulder, CO 80302, USA. [3] Department of Mineral Sciences, National Museum of Natural History, Smithsonian Institution, PO Box 37012, MRC 119Washington, D.C 20013, USA. [4] Lunar and Planetary Laboratory, University of Arizona, 1629 E University Blvd, Tucson, AZ 85721, USA. [5] Department of Geological Sciences, Jackson School of Geosciences, University of Texas, 2305 Speedway Stop C1160, Austin, TX 78712, USA. [6] Department of Earth Sciences, School of Ocean and Earth Science and Technology, University of Hawai'i at Mānoa, POST Building STE 701, 1680 East-West Road, Honolulu, HI 96822, USA. [7] INAF-Astronomical Observatory of Padova, Vic. Osservatorio 5, 35122 Padova, Italy. [8] Department of Physics, University of Central Florida, 4111 Libra Drive, Physical Sciences Bldg. 430, Orlando, FL 32816, USA. ✉email: jmolaro@psi.edu

Thermally induced breakdown or thermal stress weathering is a mechanical weathering process whereby stress fields induced by changes in temperature drive crack propagation in rock. Different thermal environments give rise to different specific fracture processes and result in different morphological features. Among these processes is thermal fatigue, sub-critical crack growth caused by diurnal thermal cycling[1], which drives progressive rock breakdown on Earth over long time periods and results in a variety of features[2–4]. One of the most common terrestrial boulder morphologies attributed to fatigue is exfoliation (surface flaking), usually in combination with other active chemical (e.g., stress corrosion, spheroidal weathering) and biogenic weathering processes[2,5,6]. At larger spatial scales, rock sheeting and dome exfoliation are typically associated with tectonic unloading, but recent works show that diurnal thermal cycling plays an important role in how these features develop[7,8]. The synergy of thermal fatigue with these other weathering mechanisms[9,10] is not well understood and disentangling their contributions to landscape evolution is challenging. Because of this, it has remained fundamentally unclear whether fatigue, in the absence of synergistic mechanisms, is able to drive mechanical weathering on Earth and, by extension, on other bodies. Many airless bodies are hypothesized to be highly susceptible to this process[11–16] due to their large diurnal temperature variations, though they lack other processes that may facilitate fatigue on Earth.

Although recent modeling and laboratory efforts have provided insight to how thermal breakdown may operate on other planetary surfaces[11,13,14], observational evidence is extremely limited. Stress fields induced in boulders from diurnal thermal cycling arise in different locations and drive crack propagation in different directions at different times of day[12]. Their complex spatiotemporal nature is tied intimately to boulder size, composition, and location, as well as the orbital and rotational properties of the body, suggesting that thermal fatigue produces a variety of morphological features on different boulders and surfaces. Molaro et al.[12] predicted that thermally driven surface disaggregation may occur in lunar boulders, but its signature is difficult to distinguish from that of micrometeorite bombardment[17,18]. Trends in spectral characteristics hint that fatigue may also drive resurfacing on asteroids[19], but definitive evidence has not been observed in spacecraft images, which are limited in both quantity and resolution. To date, the best evidence of extraterrestrial thermal fatigue is the predominant N–S trend in the orientation of boulder-scale fractures on Mars[20], which is consistent with both models[12] and terrestrial observations[3,21]. It is unknown whether this reflects weathering that occurred under the current or a past Martian climate regime, and laboratory studies show that thermally driven crack propagation is harder to achieve in anhydrous and vacuum environments than in ambient atmosphere[22–24]. Thus, although fatigue is thought of as an important driver of surface evolution, the significance of this process on airless body surfaces has remained hypothetical until now.

The OSIRIS-REx Camera Suite (OCAMS)[25] has obtained images of the surface of asteroid (101955) Bennu at pixel scales down to ~1 cm/px, providing an opportunity to search over a wide range of scales for evidence of thermal breakdown occurring in situ. Here we show observations of boulder morphologies and fractures on Bennu that are consistent with models of thermally induced rock breakdown, and not easily explained by other weathering mechanisms. Specifically, we show evidence of boulder exfoliation consistent with both terrestrial observations[2] and models[1,13] of fatigue-driven boulder degradation. These findings provide substantive and compelling evidence that thermal fracturing plays an important role on airless body surfaces, which has major implications for understanding the evolution of asteroid surfaces, orbits, and populations.

## Results

**Observations**. We used radiometrically calibrated images from the OCAMS PolyCam camera, designed to collect high resolution images of the surface[25]. Images were acquired during the Detailed Survey Baseball Diamond and Orbital B campaigns[26,27], which occurred between March 21 and July 26, 2019. The images included were taken at a spacecraft distance of 0.92–4.92 km above the surface and have pixel scales ranging from 0.9 to 6.3 cm/px (see Table 1 and "Methods" section). At the time they were acquired, these images represented the highest resolution information captured on an asteroid surface, allowing us to identify and characterize fractures and boulder surface textures at the centimeter scale. Image coverage over the polar regions is more limited and at less favorable illumination conditions, so we limited our search for weathering features to latitudes ~±70 degrees. The sizes of exfoliation layers and fractures were assessed by measuring their shadows (see "Methods" section).

**Table 1 Coordinates, attributes, and boulder and layer measurements for images shown in Figs. 1–3.**

| Figure | Lat (deg) | Lon (deg) | Diameter of boulder (m) | Layer thickness, fragment diameter, or crack width (cm) | Pixel scale (cm/px) |
|---|---|---|---|---|---|
| 1 a | −18 | 257 | 1 (cliff height) | 38.8 ± 32.6 (std = 18.6) | 6.3 |
| 1 b | 1 | 10 | 10.7 | 12.8 ± 18.9 (std = 4.0) | 3.9 |
| 1 c | 2.4 | 213.7 | 0.73 | 5.9 ± 5.6 (std = 1.6) | 0.9 |
| 1 d | −53 | 168 | 17.5 | 4.7 ± 3.7 (std = 5.5) | 6.8 |
| 1 e | 11 | 307 | 11.6 | 56.3 ± 46.4 (std = 30.5) | 6.3 |
| 1 f | 5.5 | 258 | 8 | 22.4 ± 26.4 (std = 9.7)) | 4.6 |
| 2 a | 12 | 357 | 22.2 | 140 ± 8 (average at arrows) | 3.8 |
| 2 b | −12 | 219 | 14.5 | 360 ± 8 | 3.9 |
| 2 c | 68 | 117 | 24 | 640 ± 13 | 6.6 |
| 2 d | −9 | 260 | 21.6 | 42 ± 6 | 6.1 |
| 2 e | 25 | 190 | 13.6 | 54 ± 5 | 4.7 |
| 2 f | −9 | 207.5 | 7.8 | 17 ± 5 | 4.8 |
| 3 a | −60 | 348 | 34 | – | 3.8 |
| 3 b | 26.5 | 193 | 6.2, 4.3 | – | 4.7 |
| 3 c | 5 | 93 | 18.4 | – | 3.9 |
| 3 d | 1.9 | 249 | 12 | – | 4.7 |

See "Methods" section for additional image data.

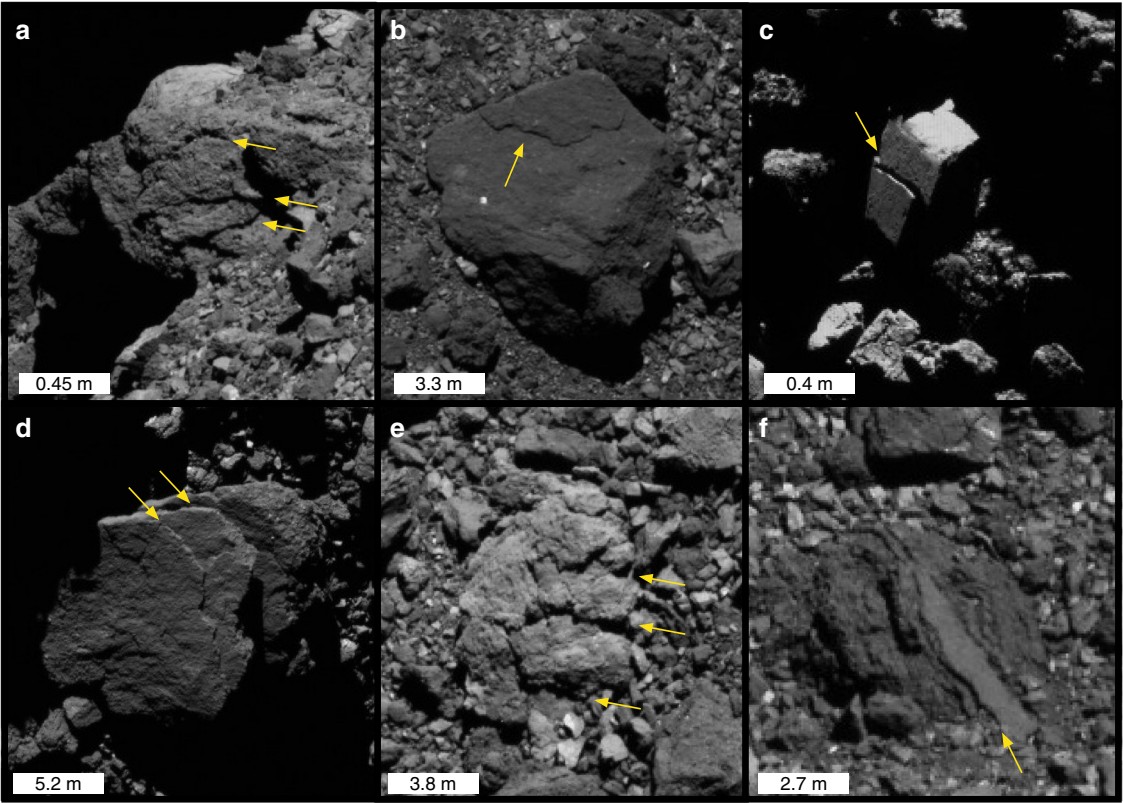

**Fig. 1 Exfoliation features observed on Bennu.** Exfoliation features on a cliff face (**a**) and on boulders (**b**–**f**) with varying size and location. Object coordinates and attributes are given in Table 1. The bright dome on the horizon of (**a**) is a boulder behind the exfoliating cliff.

Expressions of rock exfoliation (Fig. 1) are prevalent across the surface of Bennu in boulders with varying characteristics, size, and location. Bennu's boulders can be largely categorized into two types: hummocky boulders with rough surface textures and apparent brecciation, and bright, smooth boulders with angular shapes[28,29]. Exfoliation features are observed on both boulder types, and the textures and contour shapes of the flakes directly reflect their bulk characteristics. One (Fig. 1b, c) or multiple (Fig. 1a, d–f) tiered exfoliation layers may be observed parallel to a boulder's surface, and their thin edges have the roughly same orientation on a given boulder. The thickness of a given layer is relatively uniform and the surface area it occupies is typically contiguous and intact (without holes). Flakes appear to be bounded by the edges of the boulder face on which they occur, and do not occur on other faces of the same boulder. These features are observed over a range of latitudes in boulders 0.73–17.5 m diameter (Table 1), as well as on cliff-like faces (Fig. 1a). Estimating from shadows (see "Methods" section), the thickness of observed layers (in Fig. 1) ranges from centimeters to tens of centimeters (Table 1), which is consistent with terrestrial observations[6,7]. See "Modeling exfoliation" section for further discussion. Trends in feature properties with respect to boulder type, latitude, or size were not assessed due to the sample size.

Other possible expressions of thermal fatigue are in situ boulder disaggregation (Fig. 2, top) and linear fractures (Fig. 2, bottom). Many boulders appear to be breaking down in place, with disaggregated fragments visible on and around the parent boulder. Some (Fig. 2a, arrows) show apparently brighter particles and clasts at their surfaces, which may have been emplaced by particle ejection events[30] or mass movement, or could be disaggregating out of the boulder. Other boulders show disaggregation of fragments at scales larger than the apparent

breccia clasts, such as the boulder below the cavity in Fig. 2b. Fig. 2c represents a case in between, featuring a smooth contoured crack possibly following the boundary of a breccia clast, which is larger in scale than that shown in panels a or b. With a solar incidence angle of ~45 degrees and no visible topographic highs near its edge, the crack appears to be relatively deep, though the fragment has not yet disaggregated from the parent boulder. Singular and multiple-parallel linear fractures are also observed over a range of spatial scales. The most dramatic of these fractures (Fig. 2d–f) appear to bisect boulders into two fragments whose parallel fracture faces remain in close proximity. They are observed in both smooth and hummocky boulders over a wide range of scales.

Morphological expressions of weathering processes are often strongly linked to their interaction with rock fabric (layering and lineations) and texture. Layering effects are visible in many of Bennu's boulders (Fig. 3) that are similar to various jointing and fabric textures in terrestrial rocks[7]. We infer that some boulders have an intrinsic planar foliation, suggested by the presence of surface lineations and/or multiple-parallel linear surface traces (Fig. 3a, b). In other cases, abrupt changes in texture are observed within what appears to be an otherwise-competent boulder (Fig. 3c). Differential erosion rates between layers in some boulders have resulted in the protrusion of ridges from their surfaces (Fig. 3d), suggesting variability in their bulk structure due to heterogeneity in density, composition, cohesion, or the presence or relative volume of clasts. In terrestrial rocks, different fabrics develop depending on their formation mechanism and conditions. Many of Bennu's boulders contain apparent breccia clasts at scales of order 10 cm to 1 m, suggesting that impact processes played a role in their formation. Disk-integrated measurements indicate that Bennu's surface is dominated by

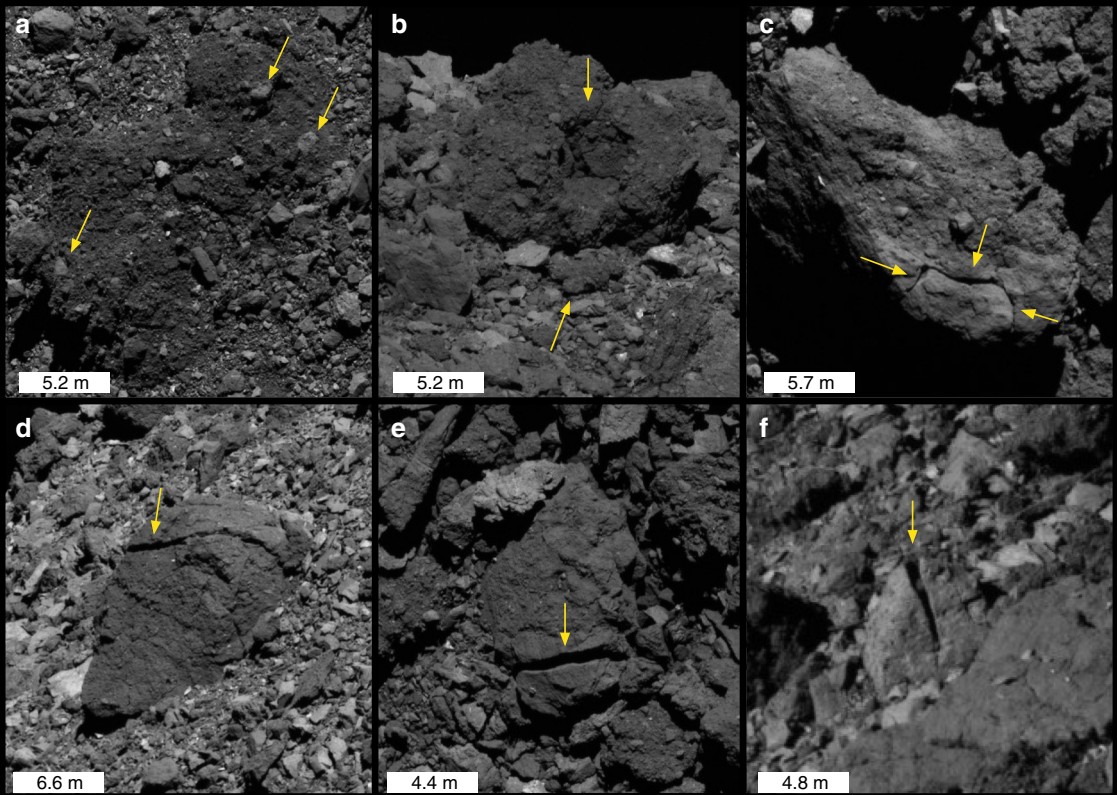

**Fig. 2 Other signs of boulder breakdown on Bennu.** Examples of disaggregation (**a**–**c**) and linear through-going fractures (**d**–**f**) in boulders of varying size. The orientation of linear fractures are in the (**d**) WNW-ESE and (**e**, **f**) N–S directions.

hydrated serpentine-group phyllosilicates[31], with the closest spectral match being CM carbonaceous chondrites. Foliation fabrics arising from phyllosilicate and chondrule shape orientations observed in CI/CM chondrites have been interpreted to result from a mixture of impact, burial, and lithification processes[32–34]. It is yet unclear what gives rise to the variety of textures observed on Bennu's boulders, emphasizing the complexity in understanding how they may interact with thermal fatigue (see "Discussion" section).

**Modeling exfoliation.** Exfoliation features (Fig. 1) are commonly observed across Bennu's surface, occur on boulders with various sizes and physical attributes, and appear to disaggregate contiguous and intact flakes of relatively uniform thickness. All of these observations are consistent with a formation mechanism that is temperature-driven rather than an impact origin. On Earth, these features develop primarily (see "Discussion" section) as a result of thermal fatigue[2]. We can demonstrate the basic mechanism that drives their formation by modeling[2,12] a boulder's thermomechanical response to the Sun. The magnitude of simulated stress fields in the boulder can be used to determine whether the threshold for crack propagation may be met, and though the model does not simulate crack propagation itself, the orientation of the stress fields informs where and at what time of day microcrack propagation will tend to occur. Following Molaro et al.[12], we performed 3D finite element simulations of diurnally induced stress fields in equatorial boulders 0.2–6 m in diameter at Bennu's perihelion (Fig. 4) using COMSOL Multiphysics. The boulders have properties consistent with terrestrial serpentinite and measurements of CI/CM chondrites (see "Methods" section), including porosities of 10% and 35%. Each boulder size is simulated using both dense (10% porosity) and porous (35% porosity) boulder properties.

Fig. 4b shows the maximum principal stress (where tensile is positive) on a cross section through a spherical boulder with diameter of 2 m at mid-morning. The Sun moves from right to left in the image. As the right side of the boulder surface heats and moves into a state of compression, a region of tension (the exfoliation region) develops in the near-surface associated with the spatial temperature gradient. The orientation of the tensile stress is normal to the boulder surface and pointing approximately in the Sun's direction. As the Sun moves overhead, the location of the exfoliation region's local maximum follows along a plane parallel to the boulder's surface, serving to drive microcrack propagation along surface-parallel planes. Over time, microcracks can coalesce into larger-scale fractures[35], leading to the development of an exfoliation flake that separates from the boulder surface. Once an exfoliation flake has begun to disaggregate, expansion and contraction of the flake itself aids in lengthening the underlying crack. As the crack grows relative to the boulder size, the rate of crack propagation increases[1]. When it nears a boundary (e.g., boulder edge or material discontinuity), it may catastrophically disrupt and disaggregate the flake. Portions of the flake may also be disaggregated prior to catastrophic disruption due to other processes (e.g., impacts) or thermal cracking at the surface[12]. Stresses in boulders that are more porous due to composition or damage accumulation are weaker in magnitude (Fig. 4), but their orientations remain unchanged. Because the stress distribution is controlled by the direction of heating, the faces on which exfoliation occurs will be influenced by boulder location with respect to the Sun.

Terrestrial observations show that one or more surface-parallel fractures may develop within the exfoliation region[3,7]. Due to the three-dimensional nature of the stress field, the depth to which the stress orientation is surface-perpendicular (the exfoliation depth) is just shallower than the depth of the local stress

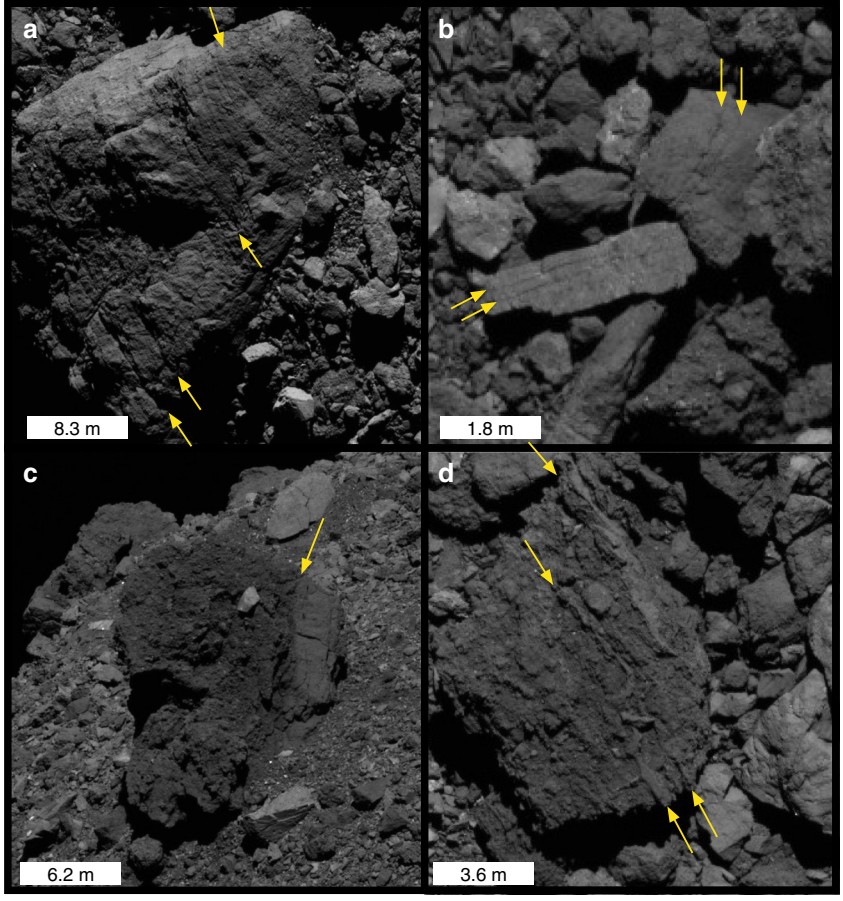

**Fig. 3 Boulder textures on Bennu.** Examples of (**a**–**d**) textures and fabrics observed in boulders of varying size.

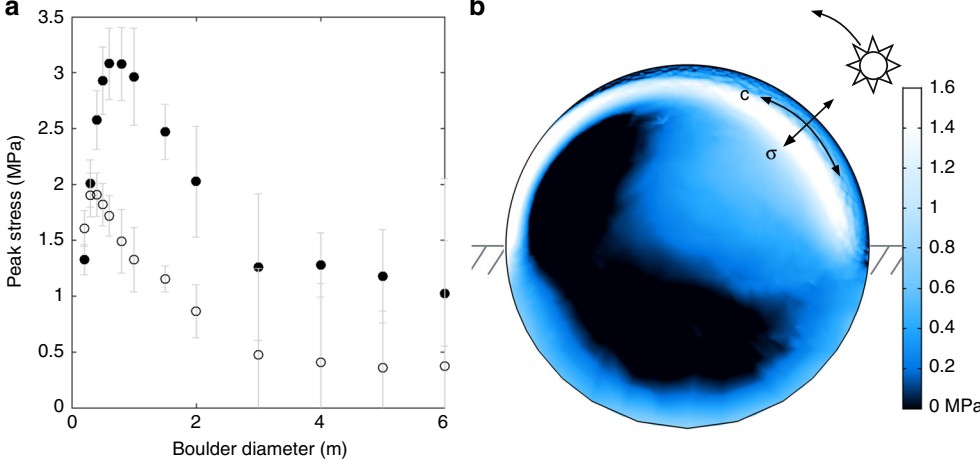

**Fig. 4 The magnitude and distribution of exfoliating stresses. a** Peak exfoliating stress in boulders of varying diameter that are dense (solid circles) and porous (open circles). Error bars represent the uncertainty due to mesh resolution and location (see "Methods" section). **b** Stress in a cross section of 2 m boulder at mid-morning. Lines show the orientation of the exfoliation stress field ($\sigma$) in the boulder's eastern near-surface and resulting direction of crack propagation (c). Black regions have negative (compressional) stress.

maximum in Fig. 4, which is controlled by both the boulder size and the diurnal thermal skin depth (~11 and 6 cm for dense and porous boulders, respectively; see "Methods" section). The exfoliation depth varies with time and is not symmetrical in all directions, but typically falls between ~1 and 3 diurnal thermal skin depths, and reaches deeper in some parts of the boulder due to interaction with other stress fields. These depths are consistent

with crack spacing in terrestrial observations[2,6,7] and with observations on Bennu (Fig. 1, Table 1). We take the magnitude of the local maximum stress as the upper limit of the exfoliating stress, which typically decreases by 20–50% at the exfoliation depth and reaches zero at the boulder surface. The upper limit ranges from ~0.3 to 3 MPa in our simulations, which is comparable to the tensile strengths of terrestrial phyllosilicate

rocks (0.5–5 MPa)[36,37] and similar soft, anisotropic materials[38,39] and exceeds estimates of the tensile strength of boulders on (162173) Ryugu (0.2 MPa)[40], which is also a carbonaceous chondrite asteroid. Further, the stresses in Fig. 4 do not constitute the highest thermal stresses that are experienced by the boulders[12], only the highest stresses thought to drive exfoliation.

Crack propagation models express stress fields in terms of the stress intensity factor, which relates the macroscopic stress field to the stress around the crack tip. The threshold to drive sub-critical crack growth in terrestrial environments typically requires a stress intensity factor that is ~10–20% of the material's fracture toughness[23]. To first order, surface-parallel microcracks can be approximated as cracks in an infinite medium, where the material's tensile strength is linearly proportional to its fracture toughness[41]. Our results show sufficient stress is present to overcome a threshold of 20% of the tensile strength, indicating that fatigue is possible on Bennu, even if these simulations overestimate stresses in real surface materials or a higher magnitude is needed to drive subcritical crack growth in vacuum.

If Bennu's materials are sufficiently weak, it is plausible that thermal shock processes[42,43], which occur when an object's tensile strength is exceeded, could potentially drive the development of exfoliation fractures instead of fatigue. On the other hand, the rubble-pile asteroid is dominated by boulders with diameters ranging from tens of centimeters to tens of meters, with finer material such as gravel and dust covering only a small portion of its visible surface[28,29]. This is not consistent with the rapid degradation and erosion rates associated with thermal shock events causing catastrophic[44] or incremental (via recurrent periods of crack growth and arrest)[23,45] rock failure, which would break down boulders quickly even over the shortest predicted timescale (100 Myr)[29] for Bennu's migration to near-Earth space. Thus, we interpret exfoliation via thermal fatigue to be the most likely origin of the observed features (Fig. 1), which is consistent with terrestrial observations[2,6,7].

## Discussion
The interaction of thermal fatigue with rock fabrics or sheeting joints may influence the direction of crack propagation due to anisotropy in material strength or the presence of joints that are mechanically weaker than adjoining layers. Fatigue cracks may deviate from their expected path if it becomes energetically favorable to propagate along a joint or weak plane[46], or propagation may occur faster in that direction. The stress field driving exfoliation moves throughout the day, so there is a high likelihood for its orientation to be favorably aligned such that it drives crack propagation along a layering plane for some portion of the day. This interaction provides a mechanism for creating planar boulder faces and fragments, and for developing single or multiple-parallel linear fractures in boulders. The weaker stresses induced at other times of day or locations within boulders may also be able to drive crack development along these planes, even if they are not associated with exfoliation features. Exfoliation and/or unrelated thermal crack propagation in layers with varying strength could explain features such as those shown in Fig. 1f, where layers appear eroded unevenly with depth, resulting in some protruding above the boulder surface or forming an overhang. The perturbation of stress fields by clasts may also serve to drive or retard crack propagation due to the mechanical strength at the clast-host boundary or to temperature gradients caused by differences in albedo or other thermal properties. Stresses induced by grain-scale effects could also play a role in crack location and distribution[11,14]. These various interactions may change the nature of fracturing among boulders, and could account for the variety of boulder morphologies found on Bennu's surface.

Annual thermal stresses may contribute to exfoliation on Bennu in a way analogous to tectonic stresses on Earth, which are not expected in a microgravity environment. The development of terrestrial sheeting joints is associated with landscape-scale compressional stresses[47], with disaggregation of surface layers aided by diurnal thermal cycling[4,8]. Because of its long orbital period, Bennu's annual thermal wave can penetrate tens of meters into the surface of even low-density solid objects. Temperature gradients set up by net heating and cooling during different times of year will induce tensile stresses below the diurnal thermal skin depth, and the superposition of annual and diurnal stresses in boulder near-surfaces may influence the spacing and propagation rates of exfoliation layers. These effects are likely to arise in boulders ~3 m in diameter (the annual thermal skin depth for porous boulders) and larger. Terrestrial sheeting joints appear as many surface-parallel cracks with characteristic spacing that is thin near the surface and increases with depth, ranging from millimeters to tens of meters apart[7]. Their formation is still not well understood, and multiple sets of joints can also become superimposed on one another as layers are eroded from the surface. These factors highlight the complexity of understanding how such features develop on Bennu's surface and predicting crack spacing.

Few mechanisms other than thermal fatigue can drive exfoliation. Chemical weathering processes such as stress corrosion and spheroidal weathering can drive exfoliation in terrestrial environments[6], but these cannot occur on bodies that lack atmospheres. The tectonic or landscape stresses that drive terrestrial exfoliation[47] at large scales are not expected in microgravity, though could have been present on the parent body. The majority of material now exposed at Bennu's surface was unlikely to have been exposed on the parent body[48], making the number of intact remnants that could have preserved such features small. Further, given the impactor flux in the main belt, the survival time for objects the size of Bennu's largest boulders is ~50 Myr, and ~1 Myr for meter scale boulders[49]. This suggests that any parent body remnants on Bennu today are likely to have accumulated enough impact damage to erode surface features, making it improbable that tectonic effects on the parent body could explain the prevalent exfoliation features we observe. It is also unlikely that the observation of exfoliation features (Fig. 1) reflects only the presence of weaknesses from planar fabrics or layering effects, as a weathering mechanism is still required to fully separate the material from boulder surfaces.

The only other mechanism that produces similar features is impact-induced spallation, where unloading of compressed material immediately following an impact may separate a shell of material from the remaining core[50,51]. Experiments on spallation in meter-sized targets show excavation of plate-like fragments resulting in a surface concavity with flat or irregularly textured floors and vertical side walls[52]. These outcomes are not consistent with observations on Bennu (Fig. 1) of exfoliating boulders that lack craters, radial fractures, or other signs of impact. The relative impactor size and velocity required to produce spallation without catastrophically disrupting the target would vary between boulders and impact experiments using carbonaceous chondrite materials are lacking, making it difficult to constrain the frequency and effects of such events. Given that the formation of the Bennu's exfoliation features can be qualitatively explained by models of thermal fatigue, and that the depths of the observed exfoliation layers are consistent with both the model and terrestrial observations, we posit that thermal fatigue is the simplest and most feasible mechanism by which these features developed on Bennu's surface.

Given the evidence that thermal fatigue is actively driving exfoliation, it is plausible that thermal stress fields at different

locations and times of day produce additional features on Bennu's surface (Fig. 2). Thermal fatigue driven by stress fields deeper in boulder interiors can drive the development of linear fractures oriented in the N–S direction[12] (Fig. 2, bottom; Table 1), such as those observed on Earth and Mars[3,20]. Without rigorous mapping of directionality, we can only report qualitatively that many fracture orientations are observed, and there are many signatures of re-shaping and material movement on Bennu that influence the alignment of these fractures over time[29,53]. Widespread signs of in situ boulder disaggregation (Fig. 2, top) may reflect the action of thermal fatigue at or near boulder surfaces[11,12,15]. Some of these features (Fig. 2a, b) are harder to ascribe solely to fatigue, as present-day impact processes (e.g., micrometeorite bombardment) and existing fracture networks from their impact histories play an important role in how the boulders break down. On the other hand, the distinctive shape of the fracture in Fig. 2c strongly suggests that it is developing progressively in situ by fatigue, as its path appears to have been controlled or influenced by adjacent clasts or boulder inhomogeneities[11,14], and the boulder does not exhibit any radial or clast-cutting fractures or fracture networks that would suggest a more energetic (e.g., impact) origin.

This work substantiates the oft-posited hypothesis that thermal fatigue is an active process on airless body surfaces. Quantifying its nature and rate on asteroids of different composition will be important in assessing how it influences the current understanding of their surface histories. Historically, the degree of space-weathering has been used as one relative measure of surface age, but both the geomorphological expression and the rate at which fatigue occurs relative to space weathering on a given body (and for a given rock type) will influence how young its surface appears to be, as well as the assigned relative ages of its craters and other surface units. Ages from cratering records will also need to be revisited, as asteroids with compositions more susceptible to thermal breakdown should experience faster rates of crater degradation and erasure. Itokawa and Bennu both have an estimated surface age of up to ~1 Gyr[29,54]; however, Itokawa's stony composition is likely to be stronger and experience slower rates of fatigue than the carbonaceous chondrite-like material of Bennu, which could make Bennu appear younger by comparison than it really is. Thermally driven resurfacing rates are estimated to be approximately three orders of magnitude higher in near-Earth space than in the main belt[19] owing to the smaller solar distance, so there may be differences in the surface properties of the two asteroid populations. A sudden increase in boulder degradation post-migration could dramatically change an asteroid's surface over a short period relative to its life in the main belt, which may contribute to the lack of small craters observed on some near-Earth asteroid surfaces[29,55].

Understanding the expression of thermal fatigue also has important implications for asteroid astronomy, as variation in composition may lead different bodies to evolve towards different states of surface roughness, thermal inertia, and optical maturity. These properties influence their spectral signatures, as well as the evolution of their orbital and rotational states due to the Yarkovsky and YORP (Yarkovsky–O'Keefe–Radzievskii–Paddack) effects. As such, thermal fatigue could directly contribute to the spectral and orbital classifications that we observe in subtle ways. For example, if fatigue preferentially and efficiently breaks down certain rock types, this may serve to homogenize the spectra of asteroids containing a wide range of volumes of that material. The disaggregation and spread of certain C-type materials across the surfaces of Bennu and Ryugu then may contribute to their similarity when viewed from afar, despite the fact that, upon visual inspection from orbit, Bennu appears to contain a much wider diversity of materials on its surface. Or, differences in asteroid orbital evolution may influence the distribution of materials throughout the Solar System, for example by preferentially aiding in the disruption and disappearance of small, low-albedo asteroids at small perihelion distances[56]. Such an effect could also influence or limit the types of materials delivered to Earth via meteoroid streams.

## Methods

**Observations and measurements**. To measure boulder sizes, images were projected onto a shape model (v28) of the asteroid using the Small Body Mapping Tool[53,57]. Image numbers and attributes are given in Table 2. Measurements of boulder sizes and boulder features (fragments and clasts in Fig. 2a–c) were obtained by drawing three-point ellipses or two-point line distances between their resolved margins. Images acquired with the spacecraft at an oblique angle to the surface do not project well over the shape model, and in these cases boulder sizes were verified by measuring the same feature in multiple images with more suitable viewing angles.

The sizes of exfoliation layers and fractures were assessed by measuring shadows on unprojected images using SAOImageDS9[58]. Image backplanes of photometric angles (emission, incidence, phase), oblique pixel scale (ground sample distance), and body-fixed geographic coordinates were calculated for each image on a per pixel basis. Calculations were performed using ray-tracing methods described by DellaGiustina et al.[27] and the v28 tessellated plate model of Bennu's shape[53]. Images were registered to the shape model using reconstructed SPICE kernels, resulting in an image-to-shape registration accuracy that varies from 5 to 20 pixels. For the six boulders shown in Fig. 1, we measured the vertical length of

---

**Table 2 Capture dates, times, and attributes for images used in this study.**

| Figure | Capture date and time (UTC) | Lat | Lon | Incidence angle | Phase angle | Azimuth angle | Pixel scale (cm/px) | Spacecraft distance (km) |
|---|---|---|---|---|---|---|---|---|
| 1 a | 20190404T22:04:58 | −18 | 257 | 50 | 39 | 9 | 6.3 | 4.5 |
| 1 b | 20190412T21:02:00 | 1 | 10 | 29 | 43.6 | −12 | 3.9 | 2.83 |
| 1 c | 20190726T23:11:44 | 2.4 | 213.7 | 82 | 85.6 | −4 | 0.9 | 0.652 |
| 1 d | 20190411T18:42:26 | −53 | 168 | 86 | 41.2 | −1 | 6.8 | 4.92 |
| 1 e | 20190404T21:22:15 | 11 | 307 | 35 | 39.5 | 23 | 6.3 | 4.59 |
| 1 f | 20190328T21:38:17 | 5.5 | 258 | 32 | 45.8 | 25 | 4.6 | 3.37 |
| 2 a | 20190405T21:44:04 | 12 | 357 | 34 | 45 | | 3.8 | 2.78 |
| 2 b | 20190412T18:24:02 | −12 | 219 | 35 | 32 | | 3.9 | 2.8 |
| 2 c | 20190404T19:06:59 | 68 | 117 | 69 | 40.8 | | 6.6 | 4.87 |
| 2 d | 20190404T21:56:21 | −9 | 260 | 34 | 38.4 | | 6.1 | 4.52 |
| 2 e | 20190321T18:44:19 | 25 | 190 | 39 | 30.2 | | 4.7 | 3.43 |
| 2 f | 20190329T19:44:29 | −9 | 207.5 | 31 | 50.6 | | 4.8 | 3.45 |
| 3 a | 20190405T21:48:27 | −60 | 348 | 72 | 41 | | 3.8 | 2.78 |
| 3 b | 20190321T18:44:19 | 26.5 | 193 | 38 | 30.2 | | 4.7 | 3.43 |
| 3 c | 20190405T20:34:48 | 5 | 93 | 35 | 41 | | 3.9 | 2.77 |
| 3 d | 20190321T18:01:09 | 1.9 | 249 | 29 | 30.5 | | 4.7 | 3.45 |

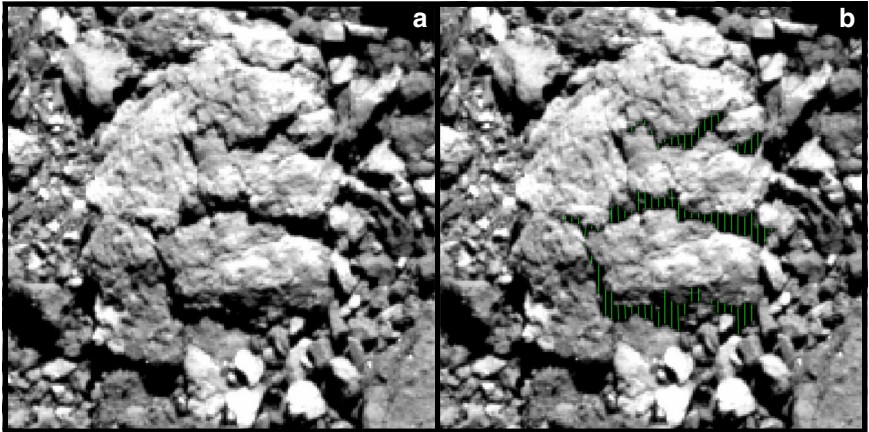

**Fig. 5 Shadow length measurements.** Example of **a** the thickness of exfoliation flakes from Fig. 1e measured by **b** their shadow lengths (green lines) captured using DS9.

each shadow by its integer number of pixels along the axis in the vertical direction in the plane of the image, taken at intervals of three horizontal pixels over its entire length (Fig. 5). These tracks were averaged to determine a vertical shadow length ($s$) for each boulder. In cases where multiple layers were visible, each layer was measured, and all measurements from all layers were averaged to determine $s$. A correction was made to convert the vertical measurement to a projected shadow length ($l$) along the axis in the direction of the Sun by multiplying $l = sp/\cos(\theta)$, where $p$ and $\theta$ are the pixel scale and solar azimuth angle (degrees east of north), respectively. To determine the height ($h$) of the layers projecting the shadows, both the position of the Sun and the spacecraft must be accounted for, giving $h = l\cos(i)/\sin(\alpha_o)$, where $i$ and $\alpha_o$ are the solar incidence and phase angles, respectively. The subscript on the phase angle is only to distinguish it from the coefficient of thermal expansion in this text. The pixel scale, incidence angle, and phase angle used for each boulder were averaged across the boulder surface, which improves uncertainty introduced by image-to-shape misregistration. The solar azimuth angle was measured directly from the shadow angles in the images.

The uncertainty for these measurements was taken to be plus or minus one vertical pixel on either end of a track and plus or minus 5 degrees in azimuth. However, to account for image misregistraton against the shape model, which determines the pixel ground sample distance and photometric angles, we factor in an additional 32% uncertainty; this corresponds to the expected error from images that are offset from the shape model by 20 pixels (e.g., worst-case image-to-shape misregistration). One additional pixel was included to account for the modest distortion in the unprojected images from PolyCam. This yielded an uncertainty value given by $(0.32s + 3)p \cos(i)/\sin(\alpha_o) \cos(\theta + 5)$ in centimeters. In Fig. 1d, the thickness is reported only for the top exfoliation layer, as limited shadowing on the layer below only yielded three measurements. Given its substantial thickness relative to the layer above it, we determined that this was not enough data points to be statistically meaningful.

The widths of the cracks in Fig. 2d–f were also measured in DS9 by the same method, except that measurements were taken at intervals of 10 pixels and no shadow correction was applied. The spatial uncertainty is given as 32%, based on the worst-case image-to-shape misregistration. In the case of Fig. 2f, measurements were taken across the entire length of the crack rather than only the shadowed region, as some material can be seen through the crack and behind the boulder.

**Model setup.** Following Molaro et al.[12], we simulated diurnal thermal stresses in equatorial spherical boulders 0.2, 0.3 0.4, 0.5, 0.6, 0.8, 1, 2, 3, 4, 5, and 6 m in diameter at Bennu's perihelion using the COMSOL Multiphysics software. Each boulder was embedded halfway within a rectangular volume of regolith (unconsolidated material or soil) such that the top of the boulder was open to space and the bottom was buried. The sides of the regolith block had fixed displacements and periodic temperature boundary conditions. The bottom of the block was also fixed, with a zero heat flux boundary condition. The surfaces of the regolith block and boulder were free to move in response to thermal forcing. The boundary between the boulder and regolith was defined as a shared "thermal contact" layer that conducts heat between two domains with different properties. This type of boundary is free to move spatially, and the heat transfer across it is $(k_{eff}/\delta)\Delta T$, where $k_{eff}$ is the effective thermal conductivity of the two materials and $\delta$ is the thickness of the boundary layer (assumed to be $10^{-6}$ m). The sensitivity of the model to these settings has a negligible impact on the results[12]. A tetrahedral mesh was applied to the model geometry.

Incident solar radiation was applied to the surface by defining a blackbody at infinite distance with a flux of 1361 W/m$^2$, with time-dependent coordinates computed using the NAIF SPICE Toolkit. The heat flux was adjusted by the solar distance, also calculated from SPICE. The coordinates were computed over a

period of one solar day at Bennu's most recent perihelion date, assuming a latitude and longitude of 0. COMSOL accounts for the local surface slope and orientation of each mesh element at the surface of the geometry, and also incorporates radiation exchange and scattering between them. We applied a correction to account for the size of the solar disc during a local sunrise or sunset by scaling the incident flux linearly with the portion of the solar disc visible above the local horizon.

The full suite of measurements of the thermal and mechanical properties of carbonaceous chondrite materials are not available in the literature. While the thermal properties of Bennu's boulders can be inferred within a parameter space from its thermal inertia, their thermal and mechanical properties are physically interwined and how the latter may vary across that parameter space is not constrained by spacecraft measurements. For this reason, it is most physically realistic to simulate boulders that have the properties of a terrestrial analog material, for which both thermal and corresponding mechanical properties are well constrained. The closest spectral match to Bennu's surface is CM carbonaceous chondrites[31], which are largely composed of serpentine-group phyllosilicates. Since the other minerals make up only a small volume of the total composition, we assumed the boulders to have the bulk properties of terrestrial serpentinite, which ranges in porosity from 10 to 35%. The latter is midrange between measurements of CM and CI chondrite meteorite porosity (23 and 35%, respectively)[59] and Bennu's bulk porosity (50%)[53]. Each boulder size was simulated using both dense (10% porosity) and porous (35% porosity) boulder properties, yielding 24 total simulations. For a porosity of 10%, the thermal and mechanical properties were a density ($\rho$) of 2510 kg/m[3,60], thermal conductivity ($k$) of 2.5 W/m K[61], linear thermal expansion coefficient ($\alpha$) of $8 \times 10^{-6}$ m$^{-1}$ [62], and Young's modulus ($E$) of 35 GPa and Poisson's ($v$) ratio of 0.34[60]. For a porosity of 35%, these values were a density of 1812 kg/m$^3$ and Young's modulus of 15 GPa and Poisson's ratio of 0.05[36]. We used a thermal conductivity of 0.5 W/m K, which is consistent with that measured for CM chondrite meteorites[63] and the linear decrease in conductivity of serpentinite expected with increased porosity[61]. All materials (including the regolith, below) had a temperature-dependent heat capacity ($c_p$) following ref. [64]. and an albedo of 0.044[28]. See ref. [12]. for additional details regarding the sensitivity of the model to material properties.

Using these material properties, Bennu's rotation period (P) of 4.288 h, and a $c_p$ value of 755 J/kg K at 300 K, we obtain a thermal skin depth $\left(\sqrt{2kP/(\pi\rho c_p)}\right)$ of 11 and 6 cm for dense and porous boulders, respectively. These are deeper than the range of 0.8 to 3 cm reported for Bennu[28], which was derived from the thermal inertia value measured for Bennu's surface of ~350 J m$^{-2}$ K$^{-1}$ s$^{-\frac{1}{2}}$ [28]. This discrepancy comes from the fact that the thermal inertia $\left(\sqrt{k\rho c_p}\right)$ of our terrestrial analog material does not match that of Bennu. The value for the dense and porous boulders in the model were 2177 and 827 J m$^{-2}$ K$^{-1}$ s$^{-\frac{1}{2}}$, respectively. Their higher values indicate that one or more of the thermal properties we use do not match Bennu's boulders, or that Bennu's measured thermal inertia reflects either boulder surface porosity due to damage accumulation or the presence of centimeter (or smaller) sized particles on the surface that serve to lower its effective thermal inertia relative to solid rock. This discrepancy has a negligible effect on the magnitude of induced thermal stresses, and therefore does not affect our conclusion that thermal fatigue occurs on Bennu. A decrease in diffusivity to better match Bennu's thermal inertia without a corresponding decrease in Young's modulus would actually increase the stress slightly. The more important effect is that it would decrease the depth to which exfoliating stresses reach, but this would not change our results at the order of magnitude level.

We addressed the discrepancy in part by ensuring that the thermal inertia of the combined boulder and surrounding material does match Bennu's surface, and therefore that the temperatures they experience are a realistic approximation. The

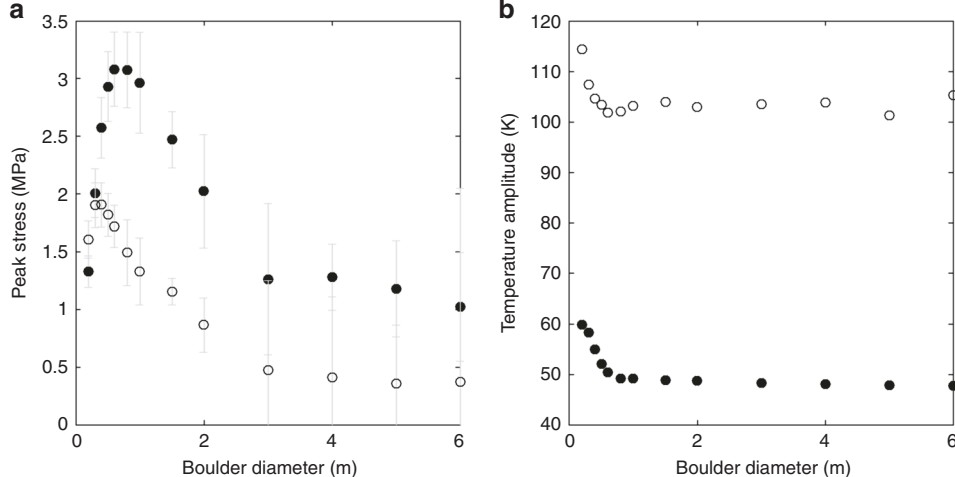

**Fig. 6 Exfoliating stresses and temperature amplitudes in boulders.** Peak exfoliating stress (**a**) and temperature amplitude (**b**) in boulders of varying diameter that are dense (solid circles) and porous (open circles). The mean temperature of all boulders was ~294 K.

density of the regolith was the bulk value for Bennu 1190 kg/m³ [53], and the thermal conductivity was adjusted such that the averaged thermal inertia of the model geometry's unburied surface area (the upper half of the sphere and surrounding regolith plane) is 350 J m⁻² K⁻¹ s⁻½ [28]. This yielded a thermal conductivity value of 0.076 W/m K and 0.125 W/m K for the regolith surrounding dense and porous boulders, respectively. While Bennu's surface is known to be rougher and denser than the lunar surface, the approximation of a boulder embedded in fine material served to increase the computational efficiency of the model. The thermal properties of the regolith are largely unimportant other than to thermally insulate the boulder's lower half and provide physical boundaries in the simulation. The regolith was assumed to have mechanical properties comparable to lunar regolith, a Young's modulus of 8 GPa[65], Poisson's ratio of 0.4[66], and linear coefficient of expansion of $2.4 \times 10^{-4}$ m⁻¹ [67]. Since unconsolidated materials have low elastic moduli and thermal expansion, this assumption eliminated any possible confining pressure that might arise were the surrounding material assumed to be solid or artificial stresses that may arise on the thermal contact layer, both of which could alter the stress field in unrealistic ways. The nature of the stress field in a boulder that thermally interacts with other nearby boulders or that has less volume that is thermally insulated would not differ qualitatively from what we present, and the quantitative sensitivity of the stress magnitude to these effects is lower than to the boulder's mechanical properties.

To calculate the temperature and stress fields, COMSOL Multiphysics solved the localized heat balance equation for heat transfer in solids over time using an implicit solver and dynamic time step. This is given by:

$$c_p \rho \left( \frac{dT}{dt} + \boldsymbol{u} \cdot \nabla T \right) + \nabla \cdot \boldsymbol{Q} = -\alpha T \sim \left( \frac{dT}{dt} + \boldsymbol{u}_{\text{trans}} \cdot \nabla S \right) \quad (1)$$

where $Q$ is the conductive and radiative heat flux, $\boldsymbol{u}_{\text{trans}}$ is the velocity vector of translational motion, $\alpha$ is the coefficient of thermal expansion, and $S$ is the second Piola-Kirchhoff stress tensor. The (:) operator is the colon, or double dot, product. The right side of Eq. (1) accounts for thermoelastic damping, where the displacement ($\boldsymbol{u}$) is a function of the Cauchy stress tensor ($s$):

$$\rho \frac{\partial^2 \boldsymbol{u}}{\partial t^2} = \boldsymbol{f} - \nabla \cdot s \quad (2)$$

where $\boldsymbol{f}$ is the volume force vector and the density is that of the deformed state. The stress tensor ($s$) is related to the elastic ($\varepsilon_{\text{el}}$) and thermal strain tensors ($\varepsilon_{\text{th}}$):

$$s = D \sim (\varepsilon_{\text{el}} - \varepsilon_{\text{th}}) = D \sim (\varepsilon_{\text{el}} - \alpha(T - T_o)) \quad (3)$$

where $T_o$ is the strain reference temperature (defined to be the average diurnal temperature of each boulder), and $D$ is a 4ᵗʰ order elasticity tensor that is a function of the Young's modulus ($E$) and Poisson's ratio ($\nu$) of the material:

$$D = \frac{E}{(1+\nu)(1-2\nu)} \begin{bmatrix} 1-\nu & \nu & \nu & 0 & 0 & 0 \\ \nu & 1-\nu & \nu & 0 & 0 & 0 \\ \nu & \nu & 1-\nu & 0 & 0 & 0 \\ 0 & 0 & 0 & \frac{1-2\nu}{\nu} & 0 & 0 \\ 0 & 0 & 0 & 0 & \frac{1-2\nu}{\nu} & 0 \\ 0 & 0 & 0 & 0 & 0 & \frac{1-2\nu}{\nu} \end{bmatrix} \quad (4)$$

Fig. 6 shows the temperature amplitude and peak stress experienced by each boulder simulated (underlying data provided in Supplementary Table 1). The

temperature amplitude (Fig. 6b) is the maximum temperature experienced within the boulder's volume throughout the day minus the minimum temperature experienced within the boulder's volume throughout the day. The stresses reported (Fig. 6a) are the maximum principal stress, where positive stress is tensile. We compare the maximum principal stress at a given location to the material's bulk tensile strength in order to assess whether crack propagation may occur. This method is appropriate for materials experiencing brittle failure, as rock does in this temperature and pressure regime, and in cases where shear stresses are negligible. Within the exfoliation region, the first and second principal stresses are oriented parallel to the surface and are equal in magnitude, while the third principal stress is lower in magnitude (usually negative, or compressional) and oriented surface-normal. Therefore, we may reasonably ignore shear stresses because the direction in which cracks will tend to propagate is primarily controlled by the first and second principal stresses. The magnitude of the maximum principal stress represents the most amount of stress available to overcome the material's tensile strength.

The peak exfoliation stresses (that is, the highest value of the maximum principal stress to occur within a given boulder's exfoliation region throughout the day) are taken from a 2D cross section of each boulder, where the cross section is taken along the axis parallel to the Sun's path. For smaller boulders (<1.5 m), at the time at which the peak exfoliation stress occurs, its value should be the peak stress anywhere in the entire boulder's volume. The uncertainty values in Fig. 4 were determined by taking the maximum stress anywhere in the boulder at the time when peak exfoliation stress occurs and subtracting the value of the peak exfoliation stress from the 2D cross section. This accounts for uncertainty in the exact location of the peak value along the axis perpendicular to the cross section. An additional 10% of the peak stress value was added to the uncertainty to account for any enhancements of stress values because of mesh resolution, which is a minor effect for boulders in this size range which have very fine meshes. This causes higher stresses to have higher uncertainty values, which is appropriate because these factors will always produce a stronger effect in simulations with more stress.

For larger boulders, two significant sources of uncertainty come into play. First, it becomes computationally expensive to simulate fine meshes in boulders whose physical domains become increasingly larger than the diurnal thermal skin depth. Thus, decreasing mesh resolution in larger boulders results in larger individual mesh elements and higher uncertainty. Second, an additional thermal stress field arises on the west side of boulders in the near-surface from a different physical effect. This stress field is still spatiotemporally dynamic but its presence is persistent in that tensile stresses that arise during sunset do not completely subside during the night and remain to some extent by the time of the next sunset. As a result, there is always some tension in that general region of the boulder, and it overlaps with the exfoliation stress field and enhances stress magnitudes in the overlapping region. This effect occurs in all boulders but is negligible for those <1.5 m in diameter for which stresses are lower magnitude. This makes the peak stress in the boulder at that time of day no longer at the location where the local maximum due solely to exfoliation would be, which is needed to maintain consistency with our measurements for smaller boulders. In these cases, the location where the exfoliation local maximum would be was determined as best as possible by eye, and the uncertainty was determined by taking the difference in stress with the peak of the second stress field. Because the latter is higher in magnitude, this uncertainty accounts for any possible reduction in the exfoliation stress due to location, or enhancement by the additional stress field or due to mesh resolution. An amount 10% of the exfoliation local maximum stress was also added to these uncertainties.

## Data availability
OCAMS images used in this work are available via the Planetary Data System (PDS) (https://sbn.psi.edu/pds/resource/orex/ocams.html)[64], except Fig. 1c, which will be available at the above URL. Data are delivered to the PDS according to the OSIRIS-REx Data Management Plan[65] available in the OSIRIS-REx PDS archive. Higher-level products such as shape models will be available in the PDS 1 year after departure from the asteroid. Modeling data for Figs. 4, 6 are provided in Supplementary Table 1.

## Code availability
The commercial software COMSOL Multiphysics (https://www.comsol.com) was used to perform the simulations for this work.

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

## Acknowledgements

We are grateful for the tremendous amount of work done by the entire OSIRIS-REx Team that made the encounter with Bennu possible and facilitated the collection of the scientific data that was used in this work. We also thank the editor for the OSIRIS-REx team, Catherine Wolner, for helping us prepare this paper for publication. This material is based upon work supported by NASA under Contract NNM10AA11C issued through the New Frontiers Program, and by Contract 80NSSC18K0239 issued through the OSIRIS-REx Participating Scientist Program. M.P. was supported for this research by the Italian Space Agency (ASI) under the ASI-INAF agreement no. 2017-37-H.0.

## Author contributions

Per the CRediT (Contributor Roles Taxonomy) system: Conceptualization: J.L.M., K.J.W., E.R.J., R.-L.B., B.R., S.R.S., M.D., R.D.H., S. J. M., M.P., H.C., A.R., W.F.B., and D.S.L. Data curation: Formal analysis: J.L.M. Funding acquisition: J.L.M., D.S.L., K.J.W., and B.R. Investigation: J.L.M., C.A.B., D.R.G., C.D.d'A., and D.N.D. Methodology: J.L.M., E.R.J., B.R., and C.D.d'A. Resources: Software: Validation: J.L.M. and D.N.D. Visuali-zation: J.L.M. and B.R. Writing—original draft: J.L.M. Writing—review & editing: J.L.M., K.J.W., E.R.J., R.-L.B., C.A.B., D.R.G., C.D.d'A., D.N.D., B.R., S.R.S., M.D., R.D.H., S.J.M., M.P., H.C., A.R., W.F.B., and D.S.L.

## Competing interests

The authors declare no competing interests.
