## [Peer Review File · Nature Communications]

Reviewers' Comments:

Reviewer #1:

Remarks to the Author:

Review of

In situ evidence of thermally induced rock breakdown widespread on Bennu's surface

By

Molaro et al

This manuscript presents new remotely sensed imaging data to describe and analyse apparent thermally induced erosion features on boulders found on the surface of the asteroid Bennu. A FEM numerical model using COMSOL multiphysics is then used to ascertain the stress field in the boulders. The results are important for many reasons but perhaps most interestingly because they permit examination of erosion features on an atmosphere free surface. This is important because interpreted thermally induced erosion features found on Earth are always prone to the complication of potentially being formed through other weathering or erosion processes. I found that the manuscript is generally well-written, novel and of great interest to the community. I think that the methods used are entirely suitable and the interpretation of data seems robust and plausible. Following some minor corrections and clarifications I believe the manuscript would be suitable for publication.

My main concerns relate to the description of the numerical model and I would like some clarification regarding the data output from the models. Specifically, it is not clear to me how many models were run and what the different results were of each model. We are only presented with one model in the manuscript (on the right of Figure 4), and the peak stresses as a function of boulder size and porosity (on the left of Figure 4), but presumably the stress distributions also change for each rotation pattern and with different material properties of the boulder. I'm not sure what are the properties of the boulder on the right of Figure 4. Also, what is the absolute temperature field of the boulders?

It is not sufficiently clear what is the mechanism invoked for producing the thermally induced rock breakdown. Are the authors proposing that thermal microcracks are generated as the boulders are heated, similar to the mechanism of heating induced thermal cracking (i.e. Friedrich, J. T., & Wong, T.-f. (1986) *Micromechanics of thermally induced cracking in three crustal rocks*. *Journal Geophysical Research*, 91, 12743–12764)? Or are thermal microcracks generated as the rock cools following a period of irradiation, similar to the mechanism of cooling induced thermal cracking (i.e. Browning, J., Meredith, P., & Gudmundsson, A. (2016) *Cooling-dominated cracking in thermally stressed volcanic rocks*. *Geophysical Research Letters*, 43(16), 8417–8425. <https://doi.org/10.1002/2016GL070532>)? Or are differential stresses generated as the boulder is hotter in certain parts than others? I think it would be good to show the absolute temperature distribution in the modelled boulders. Is it the absolute level of T that is promoting cracking or the difference in T (i.e. ΔT) which creates some kind of thermal expansion or contraction mismatch (i.e. Friedrich and Wong, 1986 or Meredith, P. G., Knight, K. S., Boon, S. A., & Wood, I. G. (2001). *The microscopic origin of thermal cracking in rocks: An investigation by simultaneous time-of-flight neutron diffraction and acoustic emission monitoring*. *Geophysical Research Letters*, 28(10), 2105–2108. <https://doi.org/10.1029/2000GL012470>)? I think it should be stated what the range of temperatures experienced by the boulders are.

I think the manuscript is partly confused by the somewhat imprecise use of terminology. For example, in Line 149 the authors refer to the 'the direction of stress and cracking' but this is confusing as it is unclear which stress the authors are referring to? What does 'direction of cracking' mean? Do they mean the propagation of the crack tip or the opening direction of cracks,

obviously the two likely have opposite directions?

Is it possible to estimate the rate of thermal erosion on the boulders?

I think COMSOL should be presented with its official name, which I think is COMSOL Multiphysics.

Regards,
John Browning

Reviewer #2:

Remarks to the Author:

I read the article by Molaro et al. with great interest, and I commend the authors on a solid piece of work. I think this paper presents compelling evidence for thermal breakdown of rocks on Bennu, and I think it should be published after only minor revisions, with one small exception.

My main criticism concerns the definitiveness of their conclusions. While I agree that the paper presents compelling evidence for thermal breakdown (that I personally believe), I do not think it is as conclusive as the authors contend. I can envision alternate scenarios for many of the lines of evidence they suggest, and while thermal breakdown provides a coherent framework to explain the observations, I think the authors should be a bit more equivocal in their conclusions (e.g., the statement in line 279 that "This work confirms").

I will discuss Figures 1 and 2 to illustrate my point. My first thought when viewing Fig. 1F was failure along layering in the boulder. (Indeed, I was unaware if there was even layering in the rocks on Bennu until I got to Fig. 3.) It is easy to envision such rock breakdown occurring from propagation of a shock wave from a micro-impact somewhere on the boulder along a layering interface. Similarly, the purported exfoliations (e.g., Fig. 1B and 1D) could arise from the complex interaction of the shape of a micro-impact eroded rock with layering in the rock. Micro-impacts could also split rocks whole (e.g., Fig. 1C and 2D-F) or even knock a large cobble in a breccia loose (e.g., Fig. 2C).

The apparent disaggregation of some boulders (Fig. 2A and 2B) provides more compelling evidence for thermal breakdown that supports the consistent framework for all the observations. (Here, I do note that Dombard et al. [Icarus 210, 2010] presented compelling evidence for thermal disaggregation of boulders on Eros, so that work should be referenced here [sentence in lines 90-91].) So while I agree with the conclusions of the paper, I think the language needs softening in places.

I continue with the minor comments, tied to line numbers:

15: as mentioned, Dombard et al. (2010) proposed thermal fatigue for breakdown of boulders on an asteroid, which precedes any of the listed references. This work built upon the ideas of Pochat et al. (Fall AGU, P23C-1289, 2009). These works should be the new references 10 and 11 in a revised manuscript. The very prescient Pochat abstract is particularly undercited in the literature.

69: insert a comma after "uniform"

Table 1: the latitude and longitude columns need a unit label (degrees, obviously)

190: for consistency, change "Figure" to "Fig."

250: where the targets of these experiments layered? If not, that could explain the differences the authors see

289: the authors contend that Itokawa's stony composition means it would experience slower rates of fatigue, but that might not be the case. As proposed in Dombard et al. (2010), the large mismatch between the thermal diffusivity of the more metal-rich matrix that holds the more stony chondrules in Eros's boulders could lead to enhancement of fatigue because the faster thermal response of the matrix could squeeze the chondrules, focusing stresses.

338: the main body of the manuscript capitalizes "Sun," while the methods section uses lower case

370: it is unclear if these simulations are fully 3-D (i.e., the "spherical boulders" in this line) or 2-D with a circular cross section (i.e., the 2D model geometry in line 435). Please clarify.

384: the method section oscillates between the present and past tense for the work that they are doing (e.g., the "are" in this sentence vs. the "was" in the previous sentence)

405: please specify that the thermal expansion coefficient is the linear coefficient

422: restrictive clause; please replace "which" with "that" (same in lines 448 and 460)

455: this equation confused me as it 1) does not explicitly identify matrices vs. scalars, and 2) does not seem to work right with the units. While the dot product of a velocity vector with the thermal gradient would yield units of temperature per time for addition to dT/dt (i.e., the parentheses on the left side of the equation), the dot product of a velocity vector with the gradient of a stress tensor would not (parentheses on right side). Please clarify.

I wish to remain anonymous.

Reviewer #3:

Remarks to the Author:

The authors present images from the surface of Bennu, showing boulders exhibiting disaggregation and linear fractures that appear to be driven by a thermal fatigue process. The authors provide arguments in support of thermal fatigue being the most probably mechanism that is leading to these failure features.

While the images are impressive, and the authors' process for discarding meteoritic bombardment as being the source of these fracture patterns is reasonable, the authors fail to justify their conclusions through an adequate model of thermal fatigue. The model that the authors present in this paper is a purely thermo-elastic model, whereas thermal fatigue is inherently a fracture mechanics exercise. It is unfitting, and unconvincing, to rely on an elastic model and claim that it explains all the different crack patterns of Figure 2, for example.

I recommend that the authors revise the modeling section or to clearly articulate the limits of their model and the extent of which the results can be extrapolated to draw conclusions for fracture and crack propagation.

Below I outline my main questions/comments on the manuscript:

1. In the abstract (and throughout the manuscript), the authors use terms such as "rock", "regolith", and "boulders" in a seemingly interchangeable manner. I recommend the authors clearly articulate what are the size scales associated with each of these terms.
2. In the abstract: The authors claim that "boulder morphologies observed on Bennu are consistent with terrestrial observations and models of fatigue-driven exfoliation". Aren't terrestrial conditions quite different from those on Bennu's surface? Are the authors suggesting, as

well, that terrestrial conditions (humidity, etc...) are insignificant when it comes to thermal fatigue fracturing of rocks?

3. Line 11-12: Related to the previous comment. Are the authors then concluding that thermal fatigue is a capable and efficient mechanism in terrestrial contexts as well?

4. Line 26: The reference number 14 that the authors mention uses an improper scaling of the results of Delbo et al. (2014, Nature), which for some reason I do not see referenced anywhere in the manuscript.

5. Line 33: While I appreciate that there is a lack of experimental data on thermal fatigue, the authors cite Ref. 19 Kirby et al. 1979, which is related to work done on Aluminum alloys. The fatigue (and crack growth behavior in general) in metal alloys has fundamentally different mechanisms than those driving cracks in brittle materials. The authors need to justify whether the same conclusion could be extrapolated to a completely different kind of material.

6. Line 35: the use of "until now" seems too ambitious. I do not see a clear proof of the "significance" of thermal fatigue on airless bodies anywhere in this paper. For instance, nowhere in the paper do the authors discuss whether these cracks are possible to grow by thermal fatigue within a reasonable amount of time with respect to their exposure on Bennu's surface.

7. Line 41: Minor comment: I suggest the authors specify which models they are referring to. The model results presented in the same paper or ones in a different reference?

8. Line 52: What is the reason for the camera resolutions (0.9 to 6.3 cm/px) being different from those on line 38 (2 to 3 cm/px)?

9. Line 67: Are the exfoliation features or the different crack directions related to the rock's position on the asteroid?

10. Line 91: Does the resolution allow us to distinguish the bright particles? I can't seem to see them well on the figures (but this may just be my screen and/or printer)

11. Figure 2: Not a comment, but more of a thumbs up: These images are remarkable.

12. Line 110: I am not sure what the authors mean by "rock fabric"?

13. Figure 3: Again, these are impressive photos. Similar question to comment #9. Are the crack directions directly related to each rock's position on the surface? Have the different cases been examined? I may have missed this point in the manuscript.

14. Line 119 (and throughout the manuscript): it is evident that not enough data exists for asteroid rocks. However, relying on guesses (and terms such as "perhaps" or "may") too much throughout the manuscript provides a false sense of reaching conclusions that are not necessarily true.

15. Figure 4: I suggest a legend be added to distinguish the two circular symbols.

16. Figure 4: Peak stress, alone, is not a sufficient or adequate metric in fracture mechanics.

17. Lines 156-168: Again, too much fracture interpretation being derived from a model that is inherently thermo-elastic.

18. Line 160: Can the authors provide evidence (or a reference) of micro-crack coalescence in thermal fatigue of rocky (or brittle) materials to form a larger macro-crack that will continue to grow in thermal fatigue? Do micro-cracks grow in thermal fatigue? How small can a crack be and still witness extension due to thermal fatigue?

19. Line 178: What about crack-tip interactions? Can't the spacing be related to the local stress fields induced by the crack tip singularities?

20. Lines 183-184 provides a dangerous comparison. The authors claim that the peak stresses are comparable to the tensile stresses. If that were the case, why wouldn't the rocks completely fracture in a brittle, structural, and non-fatigue manner? The high stresses (comparable to the tensile strengths) would seem to indicate that the rocks are not experiencing thermal fatigue (where the driving forces are typically much smaller than the material's strength values). What about long-term stress relaxation through mechanisms such as creep?

21. Paragraph starting on line 216: The authors bring up a good point with the annual thermal stresses.

22. Line 260: To be able to confidently make this claim, some temporal estimates need to be made to justify the thermal fatigue mechanism being both possible and reasonable. A bigger question that the authors seem to tiptoe around is not whether thermal fatigue is possible, but rather if it is possible, then does it happen at a fast enough "rate" to make it relevant?

23. Line 279: I do not see that this work “confirms” that thermal fatigue is (beyond a doubt) “an active process on airless body surfaces”. Timescale analysis is highly lacking.
24. The next sentences (Lines 280-283) state exactly that. How is the mechanism “confirmed” when it has not been quantified? This paper appears to present, to a large extent, a qualitative analysis
25. Lines 305-306 provide ample discussion and analysis for work not directly addressed in this paper (despite the ideas being plausible). I suggest the authors clearly identify what constitute direct conclusions from the presented work, and what are ideas that are possible but are not explored directly in this paper.
26. Line 380: Can the authors justify why the model is not very sensitive to these parameters?
27. Lines 403: Do the authors think that handling porosity on the density-level and mechanical constants level is sufficient? Would the results be different if an explicit porosity model is introduced?
28. Line 425: Again, if we were to accept that thermal fatigue occurs on Bennu (I see no thermomechanical reasoning to suggest otherwise), then to what extent does it occur? Does the model clearly predict and/or justify all the different crack paths presented in the images? What about creep and other stress relaxation mechanisms?
29. Line 471: What type of mesh elements were used? Mesh resolution is not the only metric in a finite element analysis.
30. Line 478: I do not understand why decreasing mesh resolution yields higher uncertainty? I understand that it would be inefficient to decrease the mesh resolution beyond the limits imposed by the skin depth, but it is unclear why it would result in higher uncertainty.
31. Line 478: Can the authors further explain the additional thermal stress field on the east side of the boulder?

To conclude, I believe the authors presented images that could point to a thermal fatigue-induced rock breakdown; however, the modeling piece of the paper does not sufficiently address the complexities of a thermal fatigue crack growth. Thermal fatigue, especially when involving exfoliation cracks and/or multiple cracks, is an exercise in fracture-mechanics that cannot be satisfactorily answered by means of an elastic model.

Reviewer #1:

Major Comments:

This manuscript presents new remotely sensed imaging data to describe and analyse apparent thermally induced erosion features on boulders found on the surface of the asteroid Bennu. A FEM numerical model using COMSOL multiphysics is then used to ascertain the stress field in the boulders. The results are important for many reasons but perhaps most interestingly because they permit examination of erosion features on an atmosphere free surface. This is important because interpreted thermally induced erosion features found on Earth are always prone to the complication of potentially being formed through other weathering or erosion processes. I found that the manuscript is generally well-written, novel and of great interest to the community. I think that the methods used are entirely suitable and the interpretation of data seems robust and plausible. Following some minor corrections and clarifications I believe the manuscript would be suitable for publication.

We appreciate the positive feedback from the Reviewer. The bulk of their comments (below) relate to the description of the model and limited presentation of model results. Individual responses are detailed below, which we hope have addressed their concerns.

My main concerns relate to the description of the numerical model and I would like some clarification regarding the data output from the models. Specifically, it is not clear to me how many models were run and what the different results were of each model. We are only presented with one model in the manuscript (on the right of Figure 4), and the peak stresses as a function of boulder size and porosity (on the left of Figure 4), but presumably the stress distributions also change for each rotation pattern and with different material properties of the boulder. I'm not sure what are the properties of the boulder on the right of Figure 4. Also, what is the absolute temperature field of the boulders?

We understand from the Reviewer that the manuscript was unclear regarding some of the model details, which we agree need to be clearly presented to readers. The number of simulations completed and the material properties used are described in the Methods section of the paper under Model Setup. Two simulations were performed for each of the stated boulder sizes (statement moved to lines 502-503), with each boulder simulated using the two sets of material properties. The justification for the chosen material properties and their numerical values are described in lines 534-564. We have added a clarification to this paragraph (lines 546-548) regarding the total number of simulations: "Each boulder size was simulated using both "dense" (10% porosity) and "porous" (35% porosity) boulder properties, yielding 24 total simulations." We have added a similar statement in the main text at lines 172-173, "Each boulder size is simulated using both "dense" (10% porosity) and "porous" (35% porosity) boulder properties." The readers are referred to Molaro et al. (2017) (Line 564) for additional information regarding the sensitivity of the results to other boulder compositions.

The Reviewer correctly points out that the simulation results presented in the manuscript are limited. This choice was made partly due to the typical length and format of Nature Communications articles, but largely because we did not feel that further exploration of the results would support our primary argument and the overarching goal of the study. A full exploration of the spatial and temporal nature of 3D stress fields in lunar boulders has already been done in detail by Molaro et al. (2017), which the authors discuss in the introduction and reference as a guide for this work. The qualitative shape and spatiotemporal nature of these stress fields does not change from body to body, though some of their properties may change quantitatively (e.g., their magnitude or depth). We discuss some of the more relevant aspects of the stress fields in this context in the text (Lines 211-229) but felt that additional snapshots of the stress field (e.g., Figure 4, right) would not substantively add to the manuscript. While there is more work to be done to explore the subtle differences between stress fields in lunar and asteroid boulders, such work is better suited to a longer paper. Results of that nature would not directly support our findings here, which are aimed at establishing whether or not thermal fatigue is able to occur on Bennu's surface. In this context, the most important result is quantifying the absolute magnitude of the stresses induced in Bennu's boulders, which (by comparing to the boulder's tensile strength) can reveal whether or not fatigue may occur on its surface. Secondary to this is establishing the depth to which exfoliating stresses occur, which is discussed quantitatively in the text, rather than shown via figures.

Similar to the statements above, a comparison between temperature and stress fields is detailed by Molaro et al. (2017) for lunar boulders, which we have tried to clarify can be taken qualitatively as comparison for Bennu's boulders. However, we have included an additional figure in the Methods(Supplementary Figure 2) that provides a side-by-side comparison of peak stresses and the maximum diurnal change in temperature experienced at the locations from which the stress is quantified for each boulder. We hope this will provide readers with some additional context for the simulation results (see also related comments on this below).

It is not sufficiently clear what is the mechanism invoked for producing the thermally induced rock breakdown. Are the authors proposing that thermal microcracks are generated as the boulders are heated, similar to the mechanism of heating induced thermal cracking (i.e. Friedrich and Wong, 1986)? Or are thermal microcracks generated as the rock cools following a period of irradiation, similar to the mechanism of cooling induced thermal cracking (i.e. Browning, et al., 2016)?...[continued below]

Friedrich, J. T., & Wong, T.-f. (1986) Micromechanics of thermally induced cracking in three crustal rocks. *Journal Geophysics Research*, 91, 12743–1276

Browning, J., Meredith, P., & Gudmundsson, A. (2016) Cooling-dominated cracking in thermally stressed volcanic rocks. *Geophysical Research Letters*, 43(16), 8417–8425. <https://doi.org/10.1002/2016GL070532>

The fracturing mechanism that we discuss is thermal fatigue (Lines 4-6, reworded for clarity and including a reference), “thermal fatigue, sub-critical crack growth caused by diurnal thermal cycling (Janssen, 2002), which drives progressive rock breakdown on Earth over long time periods and results in a variety of features.” Thermal fatigue differs from the effects investigated by Browning, et al. (2016) and Friedrich and Wong (1986), who focus on crack growth that occurs in a thermal shock regime. By shock we mean a regime in which the thermal stress has exceeded the tensile strength of the rock at a given spatial scale. For example, thermal stresses associated with a temperature gradient may cause catastrophic disruption of a cobble or boulder at the bulk scale, contraction cracks may develop due to surface contraction associated with a high heat flux, or microcracks may develop along grain boundaries due to mismatches in mechanical properties or grain reorientation from expansion anisotropy. These effects all occur in a shock regime, in contrast to thermal fatigue which is a sub-critical (requiring stress lower than the tensile strength, Line 231) process. We have tried to make this distinction clearer in the text with wording changes at Lines 231-243 and by including references to the studies cited by the Reviewer.

To elaborate briefly, Friedrich and Wong (1986) and Browning, et al. (2016) investigate crack growth that occurs due to grain-scale stresses caused by differential expansion along mineral grain boundaries and grain reorientation under bulk compression or tension. This phenomenon occurs when a material is subject to a new high temperature. Experiments have shown that cracking due to this effect can occur either during the heating portion of the experiment or subsequently during the cooling phase. As demonstrated by Browning et al. (2016), typically a greater number of cracks are recorded during the cooling phase when the samples are in a bulk state of tension, though this likely varies somewhat with rock type. This phenomenon may also be observed in an experiment subjecting an object to a thermal cycle in which the temperature

reaches a new maximum for the object, but typically drops off after a few cycles once it has normalized to the thermal environment. (I do not include references here, as it is clear from the Reviewer's own work cited in the comment that they are familiar with the literature on these experiments.)

Given that the simulated stresses are comparable to the tensile strength of our analog material, it is possible that effects such as this could come into play on Bennu. We might expect this to be especially relevant on surfaces that are freshly exposed by the disaggregation of material from exfoliation, impacts, or other processes. However, identifying such effects is not possible at this time, as we lack the laboratory measurements needed to quantify crack growth in carbonaceous chondrite materials, in an appropriate thermal and vacuum environment. Modeling grain-scale stresses (e.g., Molaro et al., 2015) can provide insight into these effects once we have enough data to better constrain the mineral composition, and/or once a sample of the surface materials have been returned to Earth by the OSIRIS-REx spacecraft.

[continued from above]...Or are differential stresses generated as the boulder is hotter in certain parts than others? I think it would be good to show the absolute temperature distribution in the modelled boulders. Is it the absolute level of T that is promoting cracking or the difference in T (i.e. ΔT) which creates some kind of thermal expansion or contraction mismatch (i.e. Friedrich and Wong, 1986; Meredith, et al., 2001)? I think it should be stated what the range of temperatures experienced by the boulders are.

Meredith, P. G., Knight, K. S., Boon, S. A., & Wood, I. G. (2001). The microscopic origin of thermal cracking in rocks: An investigation by simultaneous time-of-flight neutron diffraction and acoustic emission monitoring. *Geophysical Research Letters*, 28(10), 2105–2108. <https://doi.org/10.1029/2000GL012470>

The stress fields that produce exfoliation behavior, specifically, are associated with a spatial temperature gradient in the near-surface of the boulder (Lines 193-194). As alluded to in the introduction (Lines 21-31), the driving mechanism and resulting morphological features from thermal stress fields in other parts of the boulders at other times of day differ, but we refer readers to Molaro et al. (2017) for a more detailed analysis of the other effects.

The rate of fatigue crack propagation is controlled by a combination of the material properties, size of the object, and the amplitude of the temperature cycle the object experiences. In environments where chemical synergies come into play (e.g., bodies with atmospheres), sometimes the absolute temperature has also been shown to make a difference (Janssen et al., 2002), though we do not expect this to be the case on Bennu. At the Reviewer's suggestion, we have included the temperature amplitude experienced by the boulders in Supplementary Figure 2 (and the mean temperature in the figure caption) for the readers' reference.

Janssen, M., Zuidema, J., & Wanhill, R. J. H. (2002). *Fracture Mechanics*. Delft University Press.

Minor Comments:

I think the manuscript is partly confused by the somewhat imprecise use of terminology. For example, in Line 149 the authors refer to the ‘the direction of stress and cracking’ but this is confusing as it is unclear which stress the authors are referring to? What does ‘direction of cracking’ mean? Do they mean the propagation of the crack tip or the opening direction of cracks, obviously the two likely have opposite directions?

This terminology was used in attempt to make the sentence in question less cumbersome, but to mitigate confusion we have edited Line 179 to read “Lines show the orientation of the stress field (σ) and direction of crack propagation (c).” We could not find any other places in the manuscript where this confusion was repeated, though we are happy to provide further edits should more be pointed out.

“Is it possible to estimate the rate of thermal erosion on the boulders?”

It would be difficult at this stage in our research and given the state of fatigue models that may be applied to such a problem. Some researchers have attempted to adapt models such as the Paris Law to estimate crack propagation rates and boulder lifetimes on planetary surfaces. However, this type of engineering model is designed for calculating the growth of macroscopic cracks in very specific geometric scenarios with uniform, well-characterized materials, and is challenging to apply in geologic and planetary contexts. This is partly because the model parameters are empirical, and few studies have been done for materials that are geologic in nature and/or in vacuum. This causes them to have very high uncertainties (up to multiple orders of magnitude) when applied in a planetary context. They also have other limitations in the context of how well they reflect the actual type of failure that we expect to occur in boulders undergoing thermal cycling, as compared to how well they reflect machine component failure, or other engineering applications. Improvement of such models for planetary applications is ongoing in the community, but we do not feel they are reliable at this time.

We do hope to put constraints on crack propagation rates in Bennu’s boulders but feel this requires significantly more observational analysis than we can provide at this time. Additional observations that quantify exfoliation flake thickness and spatial distribution, the size and shape of disaggregated fragments, and boulder spectral characteristics across Bennu’s surface will provide considerably more context for the application of such fatigue models and more reliable interpretations of and constraints on their results. So, the bottom line is that we feel the uncertainty on such a calculation would be too high at this time to provide a meaningful result, and we hope the Reviewer will look for future work from us on this topic.

I think COMSOL should be presented with its official name, which I think is COMSOL Multiphysics.

This has been corrected in the text.

Reviewer #2:

Major Comments:

I read the article by Molaro et al. with great interest, and I commend the authors on a solid piece of work. I think this paper presents compelling evidence for thermal breakdown of rocks on Bennu, and I think it should be published after only minor revisions, with one small exception.

We appreciate the encouraging words from the Reviewer that the work is compelling.

My main criticism concerns the definitiveness of their conclusions. While I agree that the paper presents compelling evidence for thermal breakdown (that I personally believe), I do not think it is as conclusive as the authors contend. I can envision alternate scenarios for many of the lines of evidence they suggest, and while thermal breakdown provides a coherent framework to explain the observations, I think the authors should be a bit more equivocal in their conclusions (e.g., the statement in line 279 that “This work confirms”).

I will discuss Figures 1 and 2 to illustrate my point. My first thought when viewing Fig. 1F was failure along layering in the boulder. (Indeed, I was unaware if there was even layering in the rocks on Bennu until I got to Fig. 3.) It is easy to envision such rock breakdown occurring from propagation of a shock wave from a micro-impact somewhere on the boulder along a layering interface. Similarly, the purported exfoliations (e.g., Fig. 1B and 1D) could arise from the complex interaction of the shape of a micro-impact eroded rock with layering in the rock. Micro-impacts could also split rocks whole (e.g., Fig. 1C and 2D-F) or even knock a large cobble in a breccia loose (e.g., Fig. 2C).

We do agree that Figure 1f does visually appear as though layering may play a role, which we specifically note at Lines 274-276. However, our interpretation is that the most likely explanation is fatigue-driven cracking along layers, and not impact-driven. While we acknowledge that impacts occur, none of the boulders shown in this manuscript have morphological characteristics that are consistent with impact-induced spallation, so far as we could ascertain from the literature. If the Reviewer is aware of additional literature that we should review or has more specific comments regarding our discussion of impact-induced spallation at Lines 346-355, we would be happy to discuss this further.

All this being said, we acknowledge (Lines 353-355) that impact experiments on carbonaceous chondrite materials are lacking, and experiments studying impact spallation generally are

limited. As such, the end of that paragraph concludes that fatigue-driven exfoliation is the “simplest and most feasible” explanation for our observations. In this context, the Reviewer’s general point that impacts may not be completely ruled out, and specific reference to our use of the word “confirms” in the first line of the conclusion, is fair. We have softened the language in this sentence (Line 384) to read, “This work substantiates the oft-positing hypothesis that thermal fatigue is an active process on airless body surfaces.” Per the Merriam-Webster dictionary, “substantiates” means “to *give substance to*; establish by proof *or competent evidence*,” (emphasis mine) which has a similar but less absolute connotation than “confirms.” Substantiates perhaps better conveys the result of the study, which is that we have made substantial the hypothesis that thermal fatigue is an active process on an airless body surface by providing the first observational evidence that it occurs. We have laid out a very strong case for its presence by demonstrating agreement with model predictions of fatigue. We have described how our observations are *not* consistent with findings from the relevant literature on impact processes, even if impact processes may not be completely ruled out due to the need for additional research.

The apparent disaggregation of some boulders (Fig. 2A and 2B) provides more compelling evidence for thermal breakdown that supports the consistent framework for all the observations. (Here, I do note that Dombard et al. [Icarus 210, 2010] presented compelling evidence for thermal disaggregation of boulders on Eros, so that work should be referenced here [sentence in lines 90-91].) So while I agree with the conclusions of the paper, I think the language needs softening in places.

We agree that the apparent disaggregation of some boulders provides compelling evidence for thermal breakdown; however, this type of erosion is much more challenging than exfoliation to distinguish explicitly from impact erosion. It is for this reason that we have focused the manuscript on exfoliation, for which we feel we can (and do) provide an unambiguous and convincing line of evidence that points at thermal fatigue. Regarding Dombard et al. (2010), please see the related response below.

Minor Comments:

I continue with the minor comments, tied to line numbers:

15: as mentioned, Dombard et al. (2010) proposed thermal fatigue for breakdown of boulders on an asteroid, which precedes any of the listed references. This work built upon the ideas of Pochat et al. (Fall AGU, P23C-1289, 2009). These works should be the new references 10 and 11 in a revised manuscript. The very prescient Pochat abstract is particularly undercited in the literature.

We thank the Reviewer for pointing out that the manuscript did not cite Dombard et al. (2010). We are aware of this work and the omission was not by intention. Dombard et al. (2010) has provided compelling evidence for the action of thermal breakdown and played a critical role in bringing the attention of the planetary science community to the phenomena in general. We

regret that during the editing of the manuscript this citation had somehow been cut and have added it back in at Lines 16 and 374.

Regarding Pochat et al. (2009), it is the (first) author's personal policy never to cite literature that is not peer reviewed except in extremely rare cases where it is deemed critical for a specific reason. It is difficult to assess the importance or quality of Pochat's work when no resulting study was published, especially given that, as an abstract, it provides only a brief description of methods and preliminary results. For this reason, no citation of the abstract was added to the manuscript.

69: insert a comma after "uniform"

This has been corrected.

Table 1: the latitude and longitude columns need a unit label (degrees, obviously)

This has been corrected.

190: for consistency, change "Figure" to "Fig."

We have searched the full manuscript to ensure all uses of Figure and Fig. are consistent.

250: where the targets of these experiments layered? If not, that could explain the differences the authors see

We are not sure what "differences" the Reviewer is referring to here. Rock fabrics (including layering) are described at Lines 129-154 in the manuscript. The influence of layering on the fatigue process, and its ability to explain our observations, is addressed at Lines 248-281.

289: the authors contend that Itokawa's stony composition means it would experience slower rates of fatigue, but that might not be the case. As proposed in Dombard et al. (2010), the large mismatch between the thermal diffusivity of the more metal-rich matrix that holds the more stony chondrules in Eros's boulders could lead to enhancement of fatigue because the faster thermal response of the matrix could squeeze the chondrules, focusing stresses.

Mismatches between the mechanical properties of adjacent mineral grains can, indeed, generate and concentrate stresses in rocks at the mineral grain scale. As the Reviewer points out, this was suggested as a possible contributing factor to thermal breakdown on Itokawa by Dombard et al., and has been studied in detail by the (first) author in their own work (Molaro et al., 2015). Itokawa's composition as analyzed from samples from the Hayabusa mission is primarily pyroxene and plagioclase, including a small percentage of metallic inclusions (Nakamura et al.,

2011). The ~order of magnitude difference in thermal expansion between the stony and metallic components will lead to grain scale stresses that are higher than the 3-5 MPa presented for Bennu in Figure 4. However, this is not really comparing apples to apples, as stress and strength are scale-dependent. Grain scale stresses cannot be compared directly to the bulk, macroscopic stresses as presented in this work, nor can they be compared to bulk, macroscopic measurements of material tensile strengths. It is non-trivial to analyze the influence of grain scale effects on the macroscopic behavior of boulders, and such an analysis is outside the scope of this work.

We do note, however, that Bennu's materials will also have grain scale stresses and, as an exercise, a simple comparison can be made. The grain scale stress can be estimated by taking the difference in coefficient of thermal expansion (CTE) between adjacent mineral grains and multiplying it by the (average) Young's modulus and the temperature amplitude (Kranz, 1983). If we use pyroxene (see Molaro et al., 2015) as an example for Itokawa's stony composition, assume iron or nickel inclusions with a CTE of $1.3 \times 10^{-6} \text{ K}^{-1}$ (Nix and MacNair, 1941), and a temperature amplitude of 140 K to match that of Bennu, we obtain $(140 \text{ K}) * (50 \text{ GPa}) * \text{abs}(8 \times 10^{-6} - 1.3 \times 10^{-6} \text{ K}^{-1}) = \sim 47 \text{ MPa}$. Considering it has denser materials and a smaller perihelion distance, Eros likely has a smaller temperature amplitude which would lower this estimate. Bennu's surface is dominated by hydrated phyllosilicates, but magnetite has also been observed (Hamilton et al., 2019), indicating some materials also have metallic inclusions. Magnetite has a CTE of $\sim 1.1 \times 10^{-5}$ (Holcomb, 2018), yielding $(140 \text{ K}) * (35 \text{ GPa}) * \text{abs}(8 \times 10^{-6} \text{ K}^{-1} - 1.2 \times 10^{-5} \text{ K}^{-1}) = \sim 20 \text{ MPa}$. Even materials without magnetite will have multiple constituents, and with a hypothetical CTE very close to that of the matrix material (say, $6 \times 10^{-6} \text{ K}^{-1}$), Bennu's rocks could have grain scale stress of $(140 \text{ K}) * (35 \text{ GPa}) * \text{abs}(8 \times 10^{-6} \text{ K}^{-1} - 6 \times 10^{-6} \text{ K}^{-1}) = \sim 10 \text{ MPa}$.

These are approximate, back-of-the-envelope calculations, but our goal is to show that grain scale stresses between the two bodies are not likely to be as different as the Reviewer may expect. Even in a case where Bennu has no metallic inclusions, the grain scale stress estimates for both bodies are on the same order of magnitude. The strength of these materials at the grain scale is not well constrained, but given that stony materials are known to have bulk tensile strengths approximately an order of magnitude higher than phyllosilicate materials, we hold with our original statement that thermal fatigue is likely to occur more slowly on Eros.

- Hamilton, V.E., Simon, A.A., Christensen, P.R., Reuter, D.C., Clark, B.E., Barucci, M.A., Bowles, N.E., Boynton, W.V., Brucato, J.R., Cloutis, E.A. and Connolly, H.C., 2019. Evidence for widespread hydrated minerals on asteroid (101955) Bennu. *Nature astronomy*, 3(4), pp.332-340.
- Holcomb, G.R., 2019. A review of the thermal expansion of magnetite. *Materials at High Temperatures*, 36(3), pp.232-239.
- Kranz, R. L. (1983). Microcracks in rocks: a review. *Tectonophysics*, 100, 449–480.
- Nakamura, T., Noguchi, T., Tanaka, M., Zolensky, M.E., Kimura, M., Tsuchiyama, A., Nakato, A., Ogami, T., Ishida, H., Uesugi, M. and Yada, T., 2011. Itokawa dust particles: a direct link between S-type asteroids and ordinary chondrites. *Science*, 333(6046), pp.1113-1116.
- Nix, F. C., & MacNair, D. (1941). *The Thermal Expansion of Pure Metals: Copper, Gold, Aluminum, Nickel, and Iron*. *Physical Review*, 60(8), 597–605. doi:10.1103/physrev.60.597
- Molaro, J. L., Byrne, S., & Le, J. L. (2017). Thermally induced stresses in boulders on airless body surfaces, and implications for rock breakdown. *Icarus*, 294, 247–261. <http://doi.org/10.1016/j.icarus.2017.03.008>
- Molaro, J. L., Byrne, S., & Langer, S. A. (2015). Grain-scale thermoelastic stresses and spatiotemporal temperature gradients on airless bodies, implications for rock breakdown. *Journal of Geophysical Research: Planets*, 120(2), 255–277. <http://doi.org/10.1002/2014JE004729>

338: the main body of the manuscript capitalizes “Sun,” while the methods section uses lower case

This is corrected in the Methods.

370: it is unclear if these simulations are fully 3-D (i.e., the “spherical boulders” in this line) or 2-D with a circular cross section (i.e., the 2D model geometry in line 435). Please clarify.

The latter reference to which the Reviewer points was to “the model geometry’s 2D surface area,” which we can understand was confusing. We have modified this line (Lines 595-597) to read, “the averaged thermal inertia of the model geometry’s unburied surface area (the upper half of the sphere and surrounding regolith plane)...” to avoid confusion.

384: the method section oscillates between the present and past tense for the work that they are doing (e.g., the “are” in this sentence vs. the “was” in the previous sentence)

We thank the Reviewer for noticing this inconsistency and have made edits to correct it.

405: please specify that the thermal expansion coefficient is the linear coefficient

This has been added.

422: restrictive clause; please replace “which” with “that” (same in lines 448 and 460)

These have been corrected.

455: this equation confused me as it 1) does not explicitly identify matrices vs. scalars, and 2) does not seem to work right with the units. While the dot product of a velocity vector with the thermal gradient would yield units of temperature per time for addition to dT/dt (i.e., the parentheses on the left side of the equation), the dot product of a velocity vector with the gradient of a stress tensor would not (parentheses on right side). Please clarify.

We regret the confusion Eq. 1 has caused the Reviewer, the source of which was brevity on our part. Since the model is explained in detail in Molaro et al., (2017), we did not include all of the related equations in the text. We have updated the Methods to include the additional equations (Eq. 2-4) and associated description, which we hope will be more clear.

Reviewer #3:

[some of these comments are addressed out of order so that we could respond to related comments at the same time]

Major Comments:

The authors present images from the surface of Bennu, showing boulders exhibiting disaggregation and linear fractures that appear to be driven by a thermal fatigue process. The authors provide arguments in support of thermal fatigue being the most probably mechanism that is leading to these failure features.

While the images are impressive, and the authors' process for discarding meteoritic bombardment as being the source of these fracture patterns is reasonable, the authors fail to justify their conclusions through an adequate model of thermal fatigue. The model that the authors present in this paper is a purely thermo-elastic model, whereas thermal fatigue is inherently a fracture mechanics exercise. It is unfitting, and unconvincing, to rely on an elastic model and claim that it explains all the different crack patterns of Figure 2, for example.

I recommend that the authors revise the modeling section or to clearly articulate the limits of their model and the extent of which the results can be extrapolated to draw conclusions for fracture and crack propagation.

Associated numbered comments that relate to the Reviewer's major comment:

6. Line 35: the use of "until now" seems too ambitious. I do not see a clear proof of the "significance" of thermal fatigue on airless bodies anywhere in this paper. For instance, nowhere in the paper do the authors discuss whether these cracks are possible to grow by thermal fatigue within a reasonable amount of time with respect to their exposure on Bennu's surface.
22. Line 260: **To be able to confidently make this claim, some temporal estimates need to be made to justify the thermal fatigue mechanism being both possible and reasonable.** A bigger question that the authors seem to tiptoe around is not whether thermal fatigue is possible, but rather if it is possible, then does it happen at a fast enough "rate" to make it relevant?

23. Line 279: I do not see that this work “confirms” that thermal fatigue is (beyond a doubt) “an active process on airless body surfaces”. Timescale analysis is highly lacking.
24. The next sentences (Lines 280-283) state exactly that. How is the mechanism “confirmed” when it has not been quantified? This paper appears to present, to a large extent, a qualitative analysis
(see also comments 16 and 17 below, which we address separately)

To conclude, I believe the authors presented images that could point to a thermal fatigue-induced rock breakdown; however, the modeling piece of the paper does not sufficiently address the complexities of a thermal fatigue crack growth. Thermal fatigue, especially when involving exfoliation cracks and/or multiple cracks, is an exercise in fracture-mechanics that cannot be satisfactorily answered by means of an elastic model.

Reviewer 3’s Major comments focus on the type of model which we have employed in this study, which they find to be inappropriate for this application. Their comments indicate they feel that use of an actual crack propagation model would be better suited, and that making the claim that thermal fatigue is active on the surface requires quantifying crack propagation rates to assess whether it is reasonable to expect thermal fatigue on Bennu’s surface (see the bolded line in the Reviewer comment above). We respectfully disagree with the Reviewer on these points.

We argue that it is possible and reasonable for exfoliation to occur on Bennu if we observe it on the surface. The Reviewer explicitly states that our discussion of why impact processes cannot explain our observations is reasonable. Therefore, if the Reviewer agrees that the features cannot be explained by impacts, then by process of elimination, thermal fatigue is the most feasible mechanism. No rate analysis is required to substantiate this claim if fatigue provides a viable explanation, that is if the observations are consistent (which we demonstrate) with how fatigue-driven exfoliation is observed in terrestrial environments and predicted by models to occur. Indeed, the fact that we observe exfoliation on Bennu and that fatigue is the most feasible mechanism means that we will be able to, eventually, use these observations to constrain fatigue crack propagation and erosion rates for the first time on an airless body surface.

The use of thermoelastic models to identify and assess crack orientations resulting from thermal stresses and exfoliation is well established in the literature (Holzhausen, 1989; Carslaw and Jaeger, 1954; Ingersoll et al., 1948). We have performed a quantitative analysis to constrain the magnitude of thermal stresses induced in Bennu’s boulders to ensure that such stress fields are capable of driving fatigue on Bennu. While it cannot predict a crack’s exact path through a real microstructure, it can predict the direction which cracks will tend to propagate and in what regions of the object. We disagree that such a model is “unfitting” to predict fatigue-driven exfoliation on Bennu. The Reviewer acknowledges there is no thermomechanical reason why fatigue should not occur on Bennu (see comment #28); therefore, following historical precedent, such models may be used to determine in what locations and directions fatigue cracks are likely to propagate.

The Reviewer recommends that we employ a different type of model in this study, namely a model that quantifies actual crack propagation in the model geometry. This different approach is

most useful when it is necessary to quantify exact crack paths in real objects and microstructures and to quantify crack propagation rates. Such analysis is not necessary to demonstrate that fatigue is a feasible mechanism for producing the exfoliation features that we observe. We argue that this level of detail would not necessarily add much to the study, as we have only limited understanding of the properties, composition, and microstructure of Bennu's real surface materials, and laboratory studies to measure the empirical parameters needed for crack modeling are limited for relevant materials and in vacuum environments. Therefore, any constraints we could provide on crack propagation rates would have high uncertainty, and modeling detailed crack paths on an idealized spherical boulder would not tell us anything more than assessing crack propagation directions from the thermoelastic model. Further, the amount of time for which Bennu has been in near-Earth space is not well constrained, and therefore it is unclear that a rate analysis would be a convincing piece of evidence in determining the mechanism's feasibility.

Some of the issue that the Reviewer has with our analysis is related to Figure 2, which shows boulder fractures and morphologies which are not attributed to exfoliation but could possibly be fatigue-driven. The Reviewer states above that "It is unfitting...to rely on an elastic model and claim that it explains all the different crack patterns of Figure 2." However, we are not using our model in this study to explain the crack orientations in Figure 2. In our discussion of Figure 2 (Lines 361-377), we articulated that, given the evidence that fatigue drives exfoliation on Bennu, it is "plausible" that thermal fatigue produces or contributes to the features in Figure 2, but that these features are much more difficult to ascribe explicitly to fatigue. The entire paragraph is dedicated to discussing the difficulty in identifying how the features in Figure 2 develop. We do discuss some types of fatigue cracks which we do not explicitly model in this paper, which may have caused confusion, and so we have added references throughout the paragraph that point to other works which describe these additional fatigue cracks. The modeling results that we present in the Modeling Exfoliation section are *only* compared to the exfoliation features (Figure 1) that we observe, and not to the additional features shown in Figures 2 or 3. We have added references to Figures 1 and 2 at Lines 157 and 363, respectively, to try to make this more clear to readers.

We assert that our overall approach and the model we use are appropriate for the scope of this work, are consistent with and supported by previous works in the literature, and are described and justified in the main text and in the Methods of the manuscript. By comparing our observations of exfoliation on Bennu to our modeling results and to terrestrial observations, and by eliminating other mechanisms that may possibly explain these observations, we have laid out a strong line of evidence to support our conclusion that thermal fatigue is active at Bennu's surface. We believe the remarks of all three Reviewers ultimately support this conclusion.

Minor Comments:

Below I outline my main questions/comments on the manuscript:

1. In the abstract (and throughout the manuscript), the authors use terms such as "rock",

“regolith”, and “boulders” in a seemingly interchangeable manner. I recommend the authors clearly articulate what are the size scales associated with each of these terms.

We have edited these terms in the text to be consistent with boulder referring to an object made of rock, and use the term rock only to refer to a material type rather than an object with a specific size. We have added a definition for regolith (unconsolidated material or soil) in the Methods.

2. In the abstract: The authors claim that “boulder morphologies observed on Bennu are consistent with terrestrial observations and models of fatigue-driven exfoliation”. Aren’t terrestrial conditions quite different from those on Bennu’s surface? Are the authors suggesting, as well, that terrestrial conditions (humidity, etc...) are insignificant when it comes to thermal fatigue fracturing of rocks?

3. Line 11-12: Related to the previous comment. Are the authors then concluding that thermal fatigue is a capable and efficient mechanism in terrestrial contexts as well?

As discussed in the lines preceding this, fatigue-driven exfoliation is known to occur in terrestrial environments. However, fatigue is also known to be strongly influenced by the chemical environment, occurring more efficiently in more humid environments. As a result, it has been unclear whether or not fatigue could act alone, by which we mean in the absence of moisture/chemical or other synergistic weathering mechanisms. This is part of what makes this discovery so exciting, as it substantiates the hypothesis that it can occur on airless bodies, and therefore contributes to the landscape evolution of these worlds. We have edited the wording of Line 14 to try to make this more clear: “Because of this, it has remained fundamentally unclear whether fatigue, in the absence of synergistic mechanisms, is able to drive mechanical weathering on Earth and, by extension, on other bodies.”

4. Line 26: The reference number 14 that the authors mention uses an improper scaling of the results of Delbo et al. (2014, Nature), which for some reason I do not see referenced anywhere in the manuscript.

We reference the work of Basilevsky et al. (2015)* because it represents some of the best (and only) available work that attempts to unravel the signatures of micrometeorite impact erosion and thermal fatigue in the literature. Even if the Reviewer believes their analysis is inaccurate, the point of the referenced line is, specifically, that distinguishing the two processes is difficult to do and therefore we feel the citation is appropriate. Further, as discussed in detail by Molaro et al., (2017), the breakdown rates modeled by Delbo et al. (2014) are, themselves, unreliable, highlighting the need for more work on this topic and, we would argue, emphasizing the importance of this discovery.

(*We had previously referenced Basilevsky et al., (2013), but felt their 2015 study was a better example. The two studies are similar/related, so the Reviewer’s comment still more or less applies.)

5. Line 33: While I appreciate that there is a lack of experimental data on thermal fatigue, the authors cite Ref. 19 Kirby et al. 1979, which is related to work done on Aluminum alloys. The fatigue (and crack growth behavior in general) in metal alloys has fundamentally different mechanisms than those driving cracks in brittle materials. The authors need to justify whether the same conclusion could be extrapolated to a completely different kind of material.

Kirby et al. (1979) is one of a handful of relevant experiments done under vacuum conditions, but the Reviewer's point is well taken that a study on geologic materials would be more appropriate here. We have replaced this citation with that of Krokosky and Husak (1968). The implication of the associated text is unchanged.

Krokosky, E. M., & Husak, A. (1968). Strength characteristics of basalt rock in ultra-high vacuum. *Journal of Geophysical Research*, 73(6), 2237–2247. <http://doi.org/10.1029/JB073i006p02237>

7. Line 41: Minor comment: I suggest the authors specify which models they are referring to. The model results presented in the same paper or ones in a different reference?

Our observations are consistent with both our model, and those presented in previous works. We have added references to relevant previous works at this line.

8. Line 52: What is the reason for the camera resolutions (0.9 to 6.3 cm/px) being different from those on line 38 (2 to 3 cm/px)?

This inconsistency resulted from adding an image to the paper during a late stage of the writing process, when new spacecraft had become available at higher resolution than the previously selected images. We thank the Reviewer for noting this inconsistency, which has been corrected.

9. Line 67: Are the exfoliation features or the different crack directions related to the rock's position on the asteroid?

As stated on Lines 95-96, we could not assess any trends in exfoliation with respect to boulder location due to the small sample size of our observations, though we have added a note at Lines 206-208 that “Because the stress distribution is controlled by the direction of heating, the faces on which exfoliation occurs will be influenced by boulder location with respect to the sun.” The details of this effect are better explored as part of a more extensive study rather than a short paper of this nature. We hope the Reviewer will watch for future works from the authors (Molaro et al., *in review with Journal of Geophysical Research: Planets*) that address this, and related, questions. Latitudinal trends are expected in fatigue-driven linear fractures (McFadden et al., 2005; Eppes et al., 2015), but these features are not the primary focus of this manuscript and therefore discussed only briefly at Lines 35-37 and 363-365.

McFadden, L. D., Eppes, M. C., Gillespie, A. R. & Hallet, B. Physical weathering in arid landscapes due to diurnal variation in the direction of solar heating. *Geol. Soc. America. Bull.* **117**, 161 (2005).
Eppes, M. C., Willis, A., Molaro, J., Abernathy, S. & Zhou, B. Cracks in Martian boulders exhibit preferred orientations that point to solar-induced thermal stress. *Nature Communications* **6**, 6712 (2015).

10. Line 91: Does the resolution allow us to distinguish the bright particles? I can't seem to see them well on the figures (but this may just be my screen and/or printer)

They seem clear in the images to us, and we have noted a few examples on the rock surface using yellow arrows. Though, we admit that due to the general color and roughness of the rocks it is easier to see them in the digital version of the manuscript where one may zoom in to make the images larger. We applied a slight stretch to the color of the image to enhance the contrast but are hesitant to stretch them too much lest other features become harder to see.

11. Figure 2: Not a comment, but more of a thumbs up: These images are remarkable.

It's a really incredible world! Thank you, and kudos to the entire OSIRIS-Rex team for a successful mission that has enabled us to capture these images.

12. Line 110: I am not sure what the authors mean by "rock fabric"?

Added clarification to Line 130, "rock fabric (layering and lineations)"

13. Figure 3: Again, these are impressive photos. Similar question to comment #9. Are the crack directions directly related to each rock's position on the surface? Have the different cases been examined? I may have missed this point in the manuscript.

Again, we hope the Reviewer will watch for future works from the authors (Molaro et al., *in review with Journal of Geophysical Research: Planets*) that address this, but such analysis is outside the scope of the current work.

14. Line 119 (and throughout the manuscript): it is evident that not enough data exists for asteroid rocks. However, relying on guesses (and terms such as "perhaps" or "may") too much throughout the manuscript provides a false sense of reaching conclusions that are not necessarily true.

We appreciate this comment on the writing style of the manuscript. We have tried to edit the text with this in mind, but we also do not wish to convey an absolute where one is not intended. The term "may" is appropriate in sentences where something *may be* but is *not necessarily always* the case, as well as in sentences where we mean to imply a hypothetical. We use the term both ways throughout the text and, to our own reading, their difference is clear from the context.

15. Figure 4: I suggest a legend be added to distinguish the two circular symbols.

We do not wish to add additional text to a figure that is already somewhat busy. This information is provided in the figure caption.

16. Figure 4: Peak stress, alone, is not a sufficient or adequate metric in fracture mechanics.

17. Lines 156-168: Again, too much fracture interpretation being derived from a model that is inherently thermo-elastic.

We disagree with the Reviewer on these points. Use of the maximum principal stress as a metric of assessing failure (or possibility of failure) is known as Maximum Principal Stress Theory (also called Rankine's Theory). Under this theory, a material fails by fracturing when the maximum principal (that is, the most tensile) stress exceeds the ultimate tensile strength of the material, as measured in a simple tension test. This method is appropriate for brittle materials that do not plastically deform, and in cases where shear stresses are negligible. In this temperature/pressure and timescale regime, the assumption that boulders experience brittle failure is appropriate, and shear stresses may be ignored because the third principal stress (normal to the boulder surface) is much lower in magnitude (sometimes negative, or compressional) relative to the first and second principal stresses, which are equal in magnitude and parallel to the surface. Thus, the maximum principal stress at a given location in the exfoliation region represents the most amount of stress that may go towards overcoming the material's tensile strength, and the associated strain energy at that location would go towards actually propagating the crack some distance.

If we were to actually model fatigue fracture propagation, this calculation would be done using stress intensity factors (Janssen et al., 2002), which are also based on the maximum principal stress for mode I or opening/tensile cracks. The stress intensity factor in our case would be the maximum principal stress multiplied by a geometrical factor, and when used to calculate the amount of time over which the crack grows (e.g., using the Paris Law), a critical stress threshold must be achieved in order for crack growth to occur. For reasons described above, we do not delve into engineering models of crack growth for this study. Since our goal is to assess whether or not fatigue can occur (rather than the rate at which cracks propagate), we use the typical rule of thumb that fatigue crack growth will occur if the stress experienced is approximately 20% of the material's tensile strength (Atkinson, 1984).

As previously mentioned, the first and second principal stresses are oriented parallel to the boulder surface and occur quasi-statically below the sub-solar point at the boulder surface. Because the third principal stress is lower (and/or negative) in magnitude, the orientation of the first and second principal stress fields tell us the direction in which cracks would tend to propagate. (Though, we acknowledge, of course, that real rocks have heterogeneities, defects, and weaknesses that may cause variation.) We have added some additional text to the Methods (Lines 654-664) to clarify this point, but we find no theoretical issue with this approach, which

has been used by previous researchers in terrestrial studies (e.g., Holzhausen, 1989), for the specific goals of this paper.

Atkinson, B.K., 1984. Subcritical crack growth in geological materials. *Journal of Geophysical Research: Solid Earth*, 89(B6), pp.4077-4114.

Holzhausen, G. R. (1989). Origin of sheet structure, 1. Morphology and boundary conditions. *Engineering Geology*, 27(1-4), 225–278. [http://doi.org/10.1016/0013-7952\(89\)90035-5](http://doi.org/10.1016/0013-7952(89)90035-5)

Janssen, M., Zuidema, J., & Wanhill, R. J. H. (2002). *Fracture Mechanics*. Delft University Press.

18. Line 160: Can the authors provide evidence (or a reference) of micro-crack coalescence in thermal fatigue of rocky (or brittle) materials to form a larger macro-crack that will continue to grow in thermal fatigue? Do micro-cracks grow in thermal fatigue? How small can a crack be and still witness extension due to thermal fatigue?

Microcracks do grow via thermal fatigue, though the smallest or initial size of fatigue cracks is empirical and depends on the material, size and shape of the object, and other factors. The following citations have been added to Lines 5, 198, and 201 to provide additional background:

Jansen, D. P., Carlson, S. R., & Young, R. P. (1993). Ultrasonic imaging and acoustic emission monitoring of thermally induced microcracks in Lac du Bonnet granite. *Journal of Geophysical Research*, 98(B12), 22231–22243.

Janssen, M., Zuidema, J., & Wanhill, R. J. H. (2002). *Fracture Mechanics*. Delft University Press.

19. Line 178: What about crack-tip interactions? Can't the spacing be related to the local stress fields induced by the crack tip singularities?

Great question! The answer is yes but, as previously discussed, we are not modeling actual crack propagation in this study, nor are we modeling stress fields that develop on the mineral grain scale at crack tips or along grain boundaries. The way that grain scale stresses influence the rate and location of crack initiation and propagation is extremely complex, and outside the scope of this work. We do not yet have enough information about the mineral composition and structure of these materials in order to perform such an analysis, though return of the sample from the asteroid will facilitate this work down the road. We have added a comment alluding to possible grain scale effects at Line 279-280, “Stresses induced due to grain scale effects could also play a role in crack location and distribution (Molaro et al., 2015; Hazeli et al., 2018).”

20. Lines 183-184 provides a dangerous comparison. The authors claim that the peak stresses are comparable to the tensile stresses. If that were the case, why wouldn't the rocks completely fracture in a brittle, structural, and non-fatigue manner? The high stresses (comparable to the tensile strengths) would seem to indicate that the rocks are not experiencing thermal fatigue (where the driving forces are typically much smaller than the material's strength values). What about long-term stress relaxation through mechanisms such as creep?

We are simply comparing our results to measurements and reporting what we find. We find that the simulated stresses are comparable in magnitude to the tensile strengths reported for the terrestrial analog material we modeled. We do not see an issue with these lines of the text.

The higher end of the range of reported strength measurements in the literature exceeds our highest simulated stresses. Based on this finding, a reasonable interpretation is that the induced thermal stress in the boulders does not actually exceed the strength of the materials, placing it in a fatigue regime which is consistent with our observations of their morphological features. Multiple factors are also discussed which could influence the stress magnitudes: Since the stress field is a gradient (in magnitude) from the peak exfoliation stress in the near-surface decreasing to zero at the surface itself, the reported stress magnitudes are upper limits and there is a range of values which can contribute to crack propagation within the exfoliation region (Lines 234-235). Additional factors are that the material properties may differ somewhat from the analog material (discussed in the Methods), and that crack propagation in vacuum is thought to be harder than in atmosphere (Lines 40-41). A combination of factors is likely to play a role here, and further work on this topic analyzing data from multiple instruments aboard the OSIRIS-REx spacecraft will be needed in order to fully understand it. Return of the sample from Bennu's surface to Earth will also enable important research that can help address some of these questions.

Another interpretation is that the induced thermal stress in the boulders could actually exceed the strength of the materials. We note at Lines 240-243 that this suggests that thermal shock processes could play a role on the surface, but that this is not supported by the observations we present in this work.

We do not explore creep in this study, as such a process would occur over a much longer timescale than diurnal effects which we simulate here. Diurnal thermal cycling causes spatially and temporally dynamic tensile and compressional stresses to arise daily within boulders, going through a continuous cycle. The exfoliation stress fields discussed in this paper arise in the boulders starting in the morning and continuing through the afternoon, after which they subside, and the surface goes into a compressional state. A small amount of crack propagation occurs each day as this cycle continues, and the rock accumulates damage over time the drives the features we observe to develop. Creep and relaxation mechanisms act on static or quasistatic fields (long time-scale), and would not affect or interact with these short time-scale stress fields.

21. Paragraph starting on line 216: The authors bring up a good point with the annual thermal stresses.

Thank you. This will be an important consideration on many asteroid surfaces, though less important on surfaces like the Moon.

25. Lines 305-306 provide ample discussion and analysis for work not directly addressed in this paper (despite the ideas being plausible). I suggest the authors clearly identify what constitute direct conclusions from the presented work, and what are ideas that are possible but are not explored directly in this paper.

This paragraph opens with the line “Understanding the expression of thermal fatigue also has important implications for asteroid astronomy, as variation in composition may lead different bodies to evolve towards different states of surface roughness, thermal inertia, and optical maturity.” Lines 416-418 present an example of one way in which thermal fatigue could plausibly affect asteroid spectra, which is presented as an example (“For example, if fatigue...”) and not a direct conclusion of the study.

26. Line 380: Can the authors justify why the model is not very sensitive to these parameters?

We have justified this by guiding readers to Molaro et al. (2017), who performed tests to ascertain how sensitive the model is to the referenced model parameters.

27. Lines 403: Do the authors think that handling porosity on the density-level and mechanical constants level is sufficient? Would the results be different if an explicit porosity model is introduced?

We are not sure what exactly the Reviewer means by an “explicit porosity model,” but assume that they refer to a model which includes actual pore spaces in the 3D geometry. As discussed in previous responses to earlier comments from Reviewers 2 and 3, we do not model grain-scale stress fields in this study, including how such stress fields would develop due to the presence of cracks and other pore spaces. The way that grain scale stresses influence the rate and location of crack initiation and propagation is extremely complex, and outside the scope of this work. In future work, a multi-scale modeling approach that couples grain- and bulk-scale stress fields together in order to better understand their interaction will provide valuable insight on the way this process operates, but we feel it is not necessary for what we have accomplished in this manuscript.

The specific location at which a crack forms or the path a crack takes through a microstructure can be influenced by stress concentrations at cracks, pores, and grain boundaries, but the boulder-scale stress field will determine the general regions of the boulder in which cracks will form and the overall direction they will tend to propagate. The distribution and magnitude of this boulder-scale stress field, as pictured and quantified in Figure 4, would not change if we were to include microscopic pore spaces in the model geometry (were such a thing feasible in the context of computational considerations). Handling porosity through varying the boulder properties is a reasonable approach for a bulk-scale model. Porosity influences the material density, thermal conductivity, and Young’s modulus, all of which we have determined for both the “dense” and “porous” cases using measurements from the literature for terrestrial analogs and meteorite samples (Lines 548-563). These measurements are consistent with each other in both cases (by which we mean, we have measurements of each relevant parameter at each porosity), and reasonably enable the model to reproduce the behavior of the material at this scale.

28. Line 425: Again, if we were to accept that thermal fatigue occurs on Bennu (I see no

thermomechanical reasoning to suggest otherwise), then to what extent does it occur? Does the model clearly predict and/or justify all the different crack paths presented in the images? What about creep and other stress relaxation mechanisms?

The model itself does clearly predict the crack paths we describe in the paper, and we have attempted to accurately describe these predictions in the text. We feel that the description of the orientation and movement of the stress field, and the way that drives crack propagation, is expressed in a straightforward manner in the Modeling Exfoliation section of the text. If the Reviewer still feels we have not explained something clearly, perhaps they could specify more explicitly to help guide our revision. It is also unclear to us how Line 578 (Line 425 in the original submission) is related to this comment.

The questions of the “extent” that fatigue occurs (which we interpret to mean the rate) and that of creep and stress relaxation mechanisms are both addressed in earlier comments.

29. Line 471: What type of mesh elements were used? Mesh resolution is not the only metric in a finite element analysis.

Tetrahedral mesh elements were used. This has been added to Line 514 in the Methods.

30. Line 478: I do not understand why decreasing mesh resolution yields higher uncertainty? I understand that it would be inefficient to decrease the mesh resolution beyond the limits imposed by the skin depth, but it is unclear why it would result in higher uncertainty.

We are not exactly sure if we understand this comment but believe it may be related to the word “decreasing,” which the Reviewer seems to use differently than we do. We use decreasing to mean making the resolution lower, and so a mesh with lower resolution has larger elements. As mesh elements become larger relative to the spatial scale over which the changes in temperature occur, propagation of heat through the mesh becomes less continuous (more heterogeneous, stepped, or discrete) which can lead to oscillations and/or artificially high (or low) values of the temperature across the mesh (which, in turn, affects the stress value). In order to prevent the meshes from becoming too large to calculate, we must decrease the resolution of the mesh as the size of the boulder becomes larger. We have edited Lines 689-69- to read, “Thus, decreasing mesh resolution in larger boulders results in larger individual mesh elements and higher uncertainty.”

31. Line 478: Can the authors further explain the additional thermal stress field on the east side of the boulder?

Firstly, we note that this comment has pointed out a typo, as the additional thermal stress field is actually on the west side of the boulder(s). This has been corrected in the text. We have also expanded this discussion to better explain the additional stress field (Lines 690-697): “Second,

an additional thermal stress field arises on the west side of boulders in the near-surface from a different physical effect. This stress field is still spatiotemporally dynamic but its presence is persistent in that tensile stresses that arise during sunset do not completely subside during the night and remain to some extent by the time of the next sunset. As a result, there is always some tension in that general region of the boulder, and it overlaps with the exfoliation stress field and enhances stress magnitudes in the overlapping region. This effect occurs in all boulders but is negligible for those <1.5 m in diameter for which stresses are lower magnitude.”

Reviewers' Comments:

Reviewer #1:

Remarks to the Author:

I have read the revised manuscript with great interest and think that the authors have done a good job with their revisions. I hence feel that the manuscript is suitable for publication without further changes.

Reviewer #2:

Remarks to the Author:

I have reviewed the revisions made to the comments by the three reviewers, and I think the authors have responded adequately to all comments. I have no additional comments, and I look forward to the conversation continuing in the literature.

Reviewer #3:

Remarks to the Author:

The authors' reviewed manuscript constitutes an improvement from the original manuscript. However, the authors seem to have misinterpreted or misread my remarks on their computational model, and some of my questions remain unanswered or inappropriately addressed. The remarks that I expressed in the first review aimed at having the authors clearly articulate the limits of their models and what can (or cannot) be directly extrapolated from their modeling results, and how these conclusions tie in to the observations on the surface of Bennu. The authors, however, appear to be unwilling to do so.

My responses are outlined below.

Major Comments:

Reviewer 3's Major comments focus on the type of model which we have employed in this study, which they find to be inappropriate for this application. Their comments indicate they feel that use of an actual crack propagation model would be better suited, and that making the claim that thermal fatigue is active on the surface requires quantifying crack propagation rates to assess whether it is reasonable to expect thermal fatigue on Bennu's surface (see the bolded line in the Reviewer comment above). We respectfully disagree with the Reviewer on these points.

We argue that it is possible and reasonable for exfoliation to occur on Bennu if we observe it on the surface. The Reviewer explicitly states that our discussion of why impact processes cannot explain our observations is reasonable. Therefore, if the Reviewer agrees that the features cannot be explained by impacts, then by process of elimination, thermal fatigue is the most feasible mechanism. No rate analysis is required to substantiate this claim if fatigue provides a viable explanation, that is if the observations are consistent (which we demonstrate) with how fatigue-driven exfoliation is observed in terrestrial environments and predicted by models to occur. Indeed, the fact that we observe exfoliation on Bennu and that fatigue is the most feasible mechanism means that we will be able to, eventually, use these observations to constrain fatigue crack propagation and erosion rates for the first time on an airless body surface.

The use of thermoelastic models to identify and assess crack orientations resulting from thermal stresses and exfoliation is well established in the literature (Holzhausen, 1989; Carslaw and Jaeger, 1954; Ingersoll et al., 1948). We have performed a quantitative analysis to constrain the magnitude of thermal stresses induced in Bennu's boulders to ensure that such stress fields are capable of driving fatigue on Bennu. While it cannot predict a crack's exact path through a real

microstructure, it can predict the direction which cracks will tend to propagate and in what regions of the object. We disagree that such a model is “unfitting” to predict fatigue-driven exfoliation on Bennu. The Reviewer acknowledges there is no thermomechanical reason why fatigue should not occur on Bennu (see comment #28); therefore, following historical precedent, such models may be used to determine in what locations and directions fatigue cracks are likely to propagate.

The Reviewer recommends that we employ a different type of model in this study, namely a model that quantifies actual crack propagation in the model geometry. This different approach is

most useful when it is necessary to quantify exact crack paths in real objects and microstructures and to quantify crack propagation rates. Such analysis is not necessary to demonstrate that fatigue is a feasible mechanism for producing the exfoliation features that we observe. We argue that this level of detail would not necessarily add much to the study, as we have only limited understanding of the properties, composition, and microstructure of Bennu’s real surface materials, and laboratory studies to measure the empirical parameters needed for crack modeling are limited for relevant materials and in vacuum environments. Therefore, any constraints we could provide on crack propagation rates would have high uncertainty, and modeling detailed crack paths on an idealized spherical boulder would not tell us anything more than assessing crack propagation directions from the thermoelastic model. Further, the amount of time for which Bennu has been in near-Earth space is not well constrained, and therefore it is unclear that a rate analysis would be a convincing piece of evidence in determining the mechanism’s feasibility.

Some of the issue that the Reviewer has with our analysis is related to Figure 2, which shows boulder fractures and morphologies which are not attributed to exfoliation but could possibly be fatigue-driven. The Reviewer states above that “It is unfitting...to rely on an elastic model and claim that it explains all the different crack patterns of Figure 2.” However, we are not using our model in this study to explain the crack orientations in Figure 2. In our discussion of Figure 2 (Lines 361-377), we articulated that, given the evidence that fatigue drives exfoliation on Bennu, it is “plausible” that thermal fatigue produces or contributes to the features in Figure 2, but that these features are much more difficult to ascribe explicitly to fatigue. The entire paragraph is dedicated to discussing the difficulty in identifying how the features in Figure 2 develop. We do discuss some types of fatigue cracks which we do not explicitly model in this paper, which may have caused confusion, and so we have added references throughout the paragraph that point to other works which describe these additional fatigue cracks. The modeling results that we present in the Modeling Exfoliation section are only compared to the exfoliation features (Figure 1) that we observe, and not to the additional features shown in Figures 2 or 3. We have added references to Figures 1 and 2 at Lines 157 and 363, respectively, to try to make this more clear to readers.

We assert that our overall approach and the model we use are appropriate for the scope of this work, are consistent with and supported by previous works in the literature, and are described and justified in the main text and in the Methods of the manuscript. By comparing our observations of exfoliation on Bennu to our modeling results and to terrestrial observations, and by eliminating other mechanisms that may possibly explain these observations, we have laid out a strong line of evidence to support our conclusion that thermal fatigue is active at Bennu’s surface. We believe the remarks of all three Reviewers ultimately support this conclusion.

The authors seem to have understood from my comments that a change in their computational model is required.

However, I will point the authors to my original feedback: “I recommend that the authors revise the modeling section or to clearly articulate the limits of their model and the extent of which the results can be extrapolated to draw conclusions for fracture and crack propagation.”

That is, I do not think that the authors, for the purpose of this paper, must use an explicit crack growth model. However, the authors must also acknowledge that their model constitutes a ‘guide’ for their ‘best guess’ on how thermally-induced cracks ‘may’ grow (eg. line 190).

In their response, the authors state "If the Reviewer agrees that the features cannot be explained by impacts, then by process of elimination, thermal fatigue is the most feasible mechanism"; however, the authors fail to properly show that the stresses experienced within the boulder are small enough to be confident that a "fatigue" scenario is happening.

Perhaps a more fitting statement based on the authors' reasoning would be to remark that if impacts do not appear to explain these features, and since the model shows that thermally-induced stresses build up in a way that aligns with the observed cracks, then a thermal origin is likely. However, the cracks may have been formed by rapid propagation at different times of the day (due to the high tensile stresses experienced at those instances), or by slow subcritical crack growth (fatigue). The model, as it is currently, does not allow for the distinction. I expand on this point in a later paragraph as well.

The authors state that "Subcritical crack growth on Earth only requires stresses of ~20% of the material's tensile strength to occur", and acknowledge that "once a macroscale feature has developed, stresses become enhanced at the crack tip" (line 213). The authors also admit that "the upper limits" of their maximum tensile stresses are "comparable to the tensile strengths of terrestrial phyllosilicate rocks" (line 217) and "exceeds the estimate of the tensile strength of boulders on Ryugu".

The authors then conclude that thermal fatigue would be possible, even if the simulations overestimate the stresses (by up to an order of magnitude?).

First, I commend the authors on adding a reference to Atkinson (1984), which gives a good background relevant to the authors' paper.

I will note that Atkinson does not directly state that subcritical crack growth "requires stresses of ~20% of the material's tensile strength to occur", but rather "It is assumed that there exists a threshold K_{0} below which no significant crack extension can occur by stress corrosion. This is a function of the material's fracture properties and its environment. The value of K_{0} is [...] a small fraction (10-20%) of the critical stress intensity factor".

The authors' statement concerning the stress levels (and not stress intensities) and attributing the statement to Atkinson's work is, therefore, misleading, and the authors should consider rephrasing this sentence in their manuscript.

In fact, Atkinson devotes the largest part of the paper to discussing the different stress intensity factors and stress intensity thresholds. The very first equation in Atkinson's paper emphasizes that the near-field stresses around the crack are enhanced, and they make little mention of the material's "tensile strength", but rather focus on the material's "fracture toughness".

I recommend the authors refer to other supporting literature as well, such as Emmerich et al. (2007), where it is demonstrated that the relationship between fracture toughness and tensile strength is not as straight-forward as the authors seem to imply.

Emmerich, F. G. (2007). Tensile strength and fracture toughness of brittle materials. *Journal of Applied Physics*, 102(7), 073504.

Furthermore, based on the authors' response, the authors seem to think the stress intensity factor can be directly estimated from the "maximum principal stress multiplied by a geometrical factor". I assume they are basing this argument on the far-field analysis that is, for example, the subject of equation #2 in Atkinson et al. (1984)'s paper.

However, I will bring to the authors' attention that this statement is true for a remotely-applied stress field, where the stress state everywhere away from the crack tip is uniform. In the case of thermal stress, there is a local stress state that is varying in both space (stress gradients) and time along both the crack tip and the crack faces, which gives rise to a complex mixed-mode loading, and so the stress intensity factor is not easily estimated from the peak stresses alone. This, once more, highlights why the reliance on the maximum tensile stress as the only metric is not as straight-forward when discussed in the context of thermal fatigue in heterogeneous and flaw-ridden geologic materials.

I recommend the authors refer to the relevant work in literature, especially the works of Nemat-

Nasser, to cite a few:

Nemat-Nasser, S., and Oranratnachai, A. (March 1, 1979). "Minimum Spacing of Thermally Induced Cracks in Brittle Solids." *ASME. J. Energy Resour. Technol.* March 1979; 101(1): 34–40. <https://doi.org/10.1115/1.3446859>

Geyer, J. F., & Nemat-Nasser, S. (1982). Experimental investigation of thermally induced interacting cracks in brittle solids. *International Journal of Solids and Structures*, 18(4), 349-356.

The major concern voiced in my first review was that I did not agree with the authors that their work "confirms the oft-posed but previously unverified hypothesis that thermal fatigue is an active process on airless bodies", especially since the model itself does not provide much evidence to "confirm" and "verify" the hypothesis. I commend them for softening the statement to "substantiates the oft-posed hypothesis that thermal fatigue is an active process on airless body surfaces".

Since the authors do not want to perform a more detailed analysis for the purpose of this paper, I still believe that they need to dedicate a few lines to explain how we can come to the conclusion that it is thermal fatigue (and not thermal shock, for example) that is driving these cracks. Both fracture models (fatigue or shock) would be consistent with a "formation mechanism that is temperature-driven rather than an impact origin".

One may, therefore, also interpret their model's results as substantiating the hypothesis that thermal "shock" drives the crack growth on the surface, and not thermal fatigue.

Since, according to the authors' own words, the peak stresses are "comparable to the tensile strength", and "once a macroscale feature has developed, stresses become enhanced at the crack tip", then it would be quite feasible for a pre-existing crack to propagate in a stable and rapid manner through the rocks, in the direction of maximum tensile stress, all while facilitating its subsequent growth as its crack length extends.

Using the authors' same arguments, the crack path would be dictated by the profile of maximum tensile strength, just like thermal fatigue crack growth, and the distinction between thermal shock and thermal fatigue would not be easy to make.

One may also substantiate this thought process by pointing out that the peak stresses are in the range of "0.3 to 3 MPa in our simulations, which is comparable to the tensile strengths of terrestrial phyllosilicate rocks (0.5 to 5 MPa)" and "exceeds the estimate of the tensile strength of boulders on Ryugu (0.2 MPa)", and since the stress field will be concentrated at the crack tip, and further exceed the peak strength, then the cracks may grow rapidly in the direction of maximum tensile stress. The exfoliation features may then be produced as the crack grows from a high stress location to a low-stress location and then gets arrested as the surface goes into a compressional state during later times of the day. This is distinctly different from a sub-critical fatigue crack growth, but would result in similar features.

The authors seem to discard this possibility by assuring the readers that "this is not supported by the observations (Figures 1-3)", without explaining what exactly in the observations discards this possibility. I believe that the manuscript can be greatly enhanced if this distinction is made clearer.

Without indicating why thermal shock is disregarded, then the model would only imply that a "thermal origin" is likely, without determining whether it is a long-term fatigue fracture (sub-critical crack growth) or an intermittent and incremental shock fracture (repeated/cyclic dynamic crack growth and arrest) that is generating these rock features.

Note, in the discussion above I refer to "thermal shock" as related to a crack growth due to thermal stress fields that cause a stress intensity factor at the crack tip that is greater than the material's critical stress intensity factor.

In the authors' response, they state that "the higher end of the range of reported strength

measurements in the literature exceeds our highest simulated stresses.”, but this reads as if they are only considering the data that supports their hypothesis.

First, I will reiterate what the authors have acknowledged in both the text and the responses, that if there were a crack in their model, the peak stresses at the crack tip will be amplified and further exceed the tensile strength.

Second, what is the frequency distribution of the strengths for rocks of similar sizes to the ones they are studying in the models? That is, what is the probability of finding a meter-sized boulder with a tensile strength of 5 MPa (their upper range)? Also, why should we not expect that tensile strength of boulders on Bennu be comparable to the tensile strength estimates of boulders on Ryugu (0.2 MPa)?

One of the reasons why brittle geologic materials have such a large distribution of tensile strengths is due to their sub-scale heterogeneities (micro-scale flaws and cracks). The larger boulders (their meter-sized simulations) would, therefore, be more likely to be on the lower-end of the strength scale as they would have a larger distribution of flaws. See for example the strength scaling work done in literature for asteroid impact simulations:

Housen, K. R., & Holsapple, K. A. (1999). Scale effects in strength-dominated collisions of rocky asteroids. *Icarus*, 142(1), 21-33.

Holsapple, K., Giblin, I., Housen, K., Nakamura, A., & Ryan, E. (2002). Asteroid impacts: Laboratory experiments and scaling laws. *Asteroids III*, 1, 443-462.

Another paper that may be of interest to the authors is Cotto-Figueroa et al. (2016)

Cotto-Figueroa, D., Asphaug, E., Garvie, L. A., Rai, A., Johnston, J., Borkowski, L. & Morris, M. A. (2016). Scale-dependent measurements of meteorite strength: Implications for asteroid fragmentation. *Icarus*, 277, 73-77.

Realizing that this paper deals with compressive strength (although a similar conclusion can be extracted for tensile strength), the authors find that “the common use of terrestrial analog materials to predict scale-dependent strength properties significantly overestimates the strength of meter-sized asteroidal materials” and “boulders of similar composition on asteroids will have compressive strengths significantly less than typical terrestrial rocks”.

The authors should, therefore, provide the readers with enough evidence as to why they consider these features to be resulting from sub-critical thermal fatigue, as they hypothesize, and how the stress measures that they calculated from the model support their conclusions.

The authors state that “laboratory studies show that thermally driven crack propagation is harder to achieve in anhydrous and vacuum environments than in ambient atmosphere” (line 62), but the experimental procedure in the referenced paper does not appear to “thermally-drive” cracks, but rather performed compression and tensile tests, in addition to a “static fatigue” test, where according to the paper “the specimens were loaded to a stress calculated to be roughly 90% of their fracture strength and held at this stress until fracture occurred.” (that is, the stress magnitude was not varied). I suggest the authors re-phrase this sentence to properly reflect what has been tested in Krorosky’s paper, as it does not seem that Krorosky examined thermally-driven crack propagation.

I also recommend the authors refer to Atkinson et al. (1984)’s discussion on the effect of pressure on the fatigue lifetime of rocks, as this discussion is relevant to how vacuum (or the lack of pressure) may influence the fatigue thresholds.

Minor Comments:

We do not explore creep in this study, as such a process would occur over a much longer timescale than diurnal effects which we simulate here. Diurnal thermal cycling causes spatially and temporally dynamic tensile and compressional stresses to arise daily within boulders, going through a continuous cycle. The exfoliation stress fields discussed in this paper arise in the

boulders starting in the morning and continuing through the afternoon, after which they subside, and the surface goes into a compressional state. A small amount of crack propagation occurs each day as this cycle continues, and the rock accumulates damage over time the drives the features we observe to develop. Creep and relaxation mechanisms act on static or quasistatic fields (long time-scale), and would not affect or interact with these short time-scale stress fields.

I will once again quote Atkinson et al. (1984)'s paper "At stress intensity factors above the stress corrosion crack growth limit, significant crack growth will be suppressed if the stress is high enough for diffusion creep to make a significant contribution to deformation rate and thereby blunt macrocrack tips. This has been observed in ceramics."

I recommend that, in their analysis, the authors acknowledge, at the very least, that such crack-tip blunting mechanisms may occur. If such a blunting were to occur, then the cracks' fatigue growth may even be arrested. One can then make use of such a possibility to hypothesize that the features the authors are examining may have not been formed by thermal fatigue, but rather by a rapid crack growth happening at specific times of the day where the peak stresses are reached.

Formatting Comments:

There is a typo in line 73. The reference #2 needs to be a superscript.

I suggest the "ref." on line 367 to be changed to DellaGiustina et al., to be consistent with the referencing style used elsewhere (line 169, for example).

Review Response Notes:

1) In reviewing the full Reviewer response, it appears that most of the line numbers do not match with the referenced lines in the first revision document (which we downloaded from the submission system to check). For example, in the first comment below, we do not understand how Line 190 (“...estimates of the tensile strength of boulders on (162173) Ryugu...”) relates to the Reviewer’s comments on acknowledging model limitations. Later they reference a sentence in the text about stresses at crack tips as Line 213 and a sentence about crack propagation in vacuum environments as Line 62 in the first revision version of the manuscript, but which are actually lines 185 and 35, respectively. We are unsure if this was caused by some error on our part or due to accidental formatting changes on the Reviewer’s end. Regardless, we have done our best to find the lines the Reviewer intended to reference and apologize for any instances we failed to do so accurately.

2) Some of the Reviewer’s comments have been rearranged so that we could better respond to related comments at once, rather than repeating text.

Major Comment 1: Model Limitations

The authors’ reviewed manuscript constitutes an improvement from the original manuscript. However, the authors seem to have misinterpreted or misread my remarks on their computational model, and some of my questions remain unanswered or inappropriately addressed.

The remarks that I expressed in the first review aimed at having the authors clearly articulate the limits of their models and what can (or cannot) be directly extrapolated from their modeling results, and how these conclusions tie in to the observations on the surface of Bennu. The authors, however, appear to be unwilling to do so.

My responses are outlined below.

Major Comments:

[quote from our previous Review Response deleted for brevity]

The authors seem to have understood from my comments that a change in their computational model is required. However, I will point the authors to my original feedback: “I recommend that the authors revise the modeling section or to clearly articulate the limits of their model and the extent of which the results can be extrapolated to draw conclusions for fracture and crack propagation.” That is, I do not think that the authors, for the purpose of this paper, must use an explicit crack growth model. However,

the authors must also acknowledge that their model constitutes a ‘guide’ for their ‘best guess’ on how thermally-induced cracks ‘may’ grow (eg. line 190).

We appreciate the clarification from the Reviewer on their original intention. In the original feedback, the Reviewer provided (as part of the whole) the following comments (bold emphasis ours):

- **The model that the authors present in this paper is a purely thermo-elastic model, whereas thermal fatigue is inherently a fracture mechanics exercise. It is unfitting, and unconvincing,** to rely on an elastic model and claim that it explains all the different crack patterns of Figure 2, for example. I recommend that the authors revise the modeling section or to clearly articulate the limits of their model and the extent of which the results can be extrapolated to draw conclusions for fracture and crack propagation.
- **To be able to confidently make this claim, some temporal estimates need to be made to justify the thermal fatigue mechanism being both possible and reasonable.** A bigger question that the authors seem to tiptoe around is not whether thermal fatigue is possible, but rather if it is possible, then does it happen at a fast enough “rate” to make it relevant?
- To conclude, I believe the authors presented images that could point to a thermal fatigue-induced rock breakdown; however, the modeling piece of the paper does not sufficiently address the complexities of a thermal fatigue crack growth. **Thermal fatigue, especially when involving exfoliation cracks and/or multiple cracks, is an exercise in fracture-mechanics that cannot be satisfactorily answered by means of an elastic model.**
- Again, **too much fracture interpretation being derived from a model that is inherently thermo-elastic.**
- Again, if we were to accept that thermal fatigue occurs on Bennu (I see no thermomechanical reasoning to suggest otherwise), then to what extent does it occur? **Does the model clearly predict and/or justify all the different crack paths presented in the images?**

Our interpretation of the Reviewer’s explicit statements above (particularly the bold portions) was an overall focus and emphasis on the idea that our model is inadequate to explain the exfoliation behavior of the boulders, in the context of both the location and direction of crack propagation and the ability for crack propagation to occur. We interpreted these comments to be expressing skepticism of the overall paper methodology, rather than a request for additional explanation regarding the model limitations (which is a great suggestion!). The Reviewer explicitly stated more than once, and in more than one way, that our model was unfit and inadequate, and therefore when they offered we could “...clearly articulate the limits of their model and the extent of which the results can be extrapolated to draw conclusions for fracture and crack propagation,” it did not seem as though there were any limitations to the model that would be acceptable, such that the Reviewer would have been satisfied without replacing or supplementing it with a crack propagation model. As such, we focused our efforts in the Reviewer Response (and the associated revisions) on justifying the use of the model and explaining why additional modeling of crack propagation was not necessary. These efforts seem to have been satisfactory to the Reviewer, who has now stated they “do not think that the authors, for the purpose of this paper, must use an explicit crack growth model.”

We make these remarks only to emphasize that we are not unwilling to describe the model limitations in the manuscript, we simply did not understand from the Reviewer that that was the nature of their concerns or believe that doing so would satisfy said concerns. The sentiment interpreted from the Reviewer's comments above (whether intentional or not), was that the "limitation" of the model was that it should not be used. We endeavored to respond genuinely with explanation and supporting references to justify our method, which we genuinely believed would address the Reviewer's comments. We hope that the Reviewer will consider their interpretation of our second response here in this light.

Overall, the Reviewer's suggestion to expand on the model limitations is an excellent point! They suggested that we "acknowledge that their model constitutes a 'guide' for their 'best guess' on how thermally-induced cracks 'may' grow (e.g. line 190)." We are unsure which statement the referenced Line number should be attached to (see note above regarding line number mismatches), but based on nearby Line number mismatches it could be the statement "...serving to drive microcrack propagation along surface-parallel planes." We intentionally phrased the sentence this way to avoid making a more absolute statement that the stress field would propagate a crack in an exact or specific location. This seems to be the overall sentiment the Reviewer is looking for in acknowledging the model's limitations, but we can see that a more explicit statement would have been more clear. We have added the following statement to the opening paragraph of the modeling section (Lines 140-143): "The magnitude of simulated stress fields in the boulder can be used to determine whether the threshold for crack propagation may be met, and though the model does not simulate crack propagation itself, the orientation of the stress fields informs where and at what time of day microcrack propagation will tend to occur."

Major Comment 2a: Fatigue vs Shock

The following comments from the current Review all focus on different aspects of a single issue: the concept that we have not adequately justified why we interpret thermal fatigue to be the crack mechanism at play, rather than thermal shock. Their primary questions/arguments on this topic appear to focus around a few overlapping themes, including the stress magnitudes being comparable to the rock strength, value of the rock strengths themselves, and lack of clarity regarding why we dismiss shock as the relevant driving mechanism at play. We have removed some paragraphs of Reviewer text (indicated by ellipses) that were better addressed as separate issues (copying them into Major Comments 2b to 2d), such that the remaining text here (Major Comment 2a) focuses only on the question of fatigue versus shock. We respond "in-line" to some individual statements, and the remaining in bulk at the end of the section for Major Comment 2a.

In their response, the authors state "If the Reviewer agrees that the features cannot be explained by impacts, then by process of elimination, thermal fatigue is the most feasible mechanism"; however, the authors fail to properly show that the stresses experienced within the boulder are small enough to be confident that a "fatigue" scenario is happening.

In the previous review, the Reviewer had commented, “To be able to confidently make this claim [that thermal fatigue is occurring], some temporal estimates need to be made to justify the thermal fatigue mechanism being both possible and reasonable. A bigger question that the authors seem to tiptoe around is not whether thermal fatigue is possible, but rather if it is possible, then does it happen at a fast enough “rate” to make it relevant?” This previous concern that fatigue may not occur rapidly enough to be relevant indicates a concern that stresses may be too low (in a general sense, crack propagation rates increase with increased stress). If we understand the Reviewer’s new statement above correctly, they have also raised the concern that the stresses may be too high, such that thermal shock– an extremely rapid process, by comparison– is equally likely to be occurring. Generally, this all speaks to a need to clarify why we believe that thermal fatigue is the most likely case, which we do both here and in the revised version of the manuscript.

Perhaps a more fitting statement based on the authors’ reasoning would be to remark that if impacts do not appear to explain these features, and since the model shows that thermally-induced stresses build up in a way that aligns with the observed cracks, then a thermal origin is likely. However, the cracks may have been formed by rapid propagation at different times of the day (due to the high tensile stresses experienced at those instances), or by slow subcritical crack growth (fatigue). The model, as it is currently, does not allow for the distinction. I expand on this point in a later paragraph as well.

The authors state that “Subcritical crack growth on Earth only requires stresses of ~20% of the material’s tensile strength to occur”, and acknowledge that “once a macroscale feature has developed, stresses become enhanced at the crack tip” (line 213). The authors also admit that “the upper limits” of their maximum tensile stresses are “comparable to the tensile strengths of terrestrial phyllosilicate rocks” (line 217) and “exceeds the estimate of the tensile strength of boulders on Ryugu”.

The authors then conclude that thermal fatigue would be possible, even if the simulations overestimate the stresses (by up to an order of magnitude?).

...[text moved to Major Comments 2b and 2c]

Since the authors do not want to perform a more detailed analysis for the purpose of this paper, I still believe that they need to dedicate a few lines to explain how we can come to the conclusion that it is thermal fatigue (and not thermal shock, for example) that is driving these cracks. Both fracture models (fatigue or shock) would be consistent with a “formation mechanism that is temperature-driven rather than an impact origin”. One may, therefore, also interpret their model’s results as substantiating the hypothesis that thermal “shock” drives the crack growth on the surface, and not thermal fatigue. Since, according to the authors’ own words, the peak stresses are “comparable to the tensile strength”, and “once a macroscale feature has developed, stresses become enhanced at the crack tip”, then it would be quite feasible for a pre-existing crack to propagate in a stable and rapid manner through the rocks, in the direction of maximum tensile stress, all while facilitating its subsequent growth as its crack length extends.

Using the authors' same arguments, the crack path would be dictated by the profile of maximum tensile strength, just like thermal fatigue crack growth, and the distinction between thermal shock and thermal fatigue would not be easy to make.

One may also substantiate this thought process by pointing out that the peak stresses are in the range of “0.3 to 3 MPa in our simulations, which is comparable to the tensile strengths of terrestrial phyllosilicate rocks (0.5 to 5 MPa)” and “exceeds the estimate of the tensile strength of boulders on Ryugu (0.2 MPa)”, and since the stress field will be concentrated at the crack tip, and further exceed the peak strength, then the cracks may grow rapidly in the direction of maximum tensile stress. The exfoliation features may then be produced as the crack grows from a high stress location to a low-stress location and then gets arrested as the surface goes into a compressional state during later times of the day. This is distinctly different from a sub-critical fatigue crack growth, but would result in similar features. The authors seem to discard this possibility by assuring the readers that ‘this is not supported by the observations (Figures 1-3)’, without explaining what exactly in the observations discards this possibility. I believe that the manuscript can be greatly enhanced if this distinction is made clearer.

As the Reviewer has noted, we dismissed the contribution of thermal shock in only a brief statement and had not thought through the possibility of cracks growing incrementally via thermal shock in detail. After reviewing the literature, we still find this scenario unlikely (see below), but we appreciate the Reviewer bringing the possibility to our attention.

Without indicating why thermal shock is disregarded, then the model would only imply that a “thermal origin” is likely, without determining whether it is a long-term fatigue fracture (sub-critical crack growth) or an intermittent and incremental shock fracture (repeated/cyclic dynamic crack growth and arrest) that is generating these rock features.

Note, in the discussion above I refer to “thermal shock” as related to a crack growth due to thermal stress fields that cause a stress intensity factor at the crack tip that is greater than the material's critical stress intensity factor.

...[Text moved to Major Comment 2d]

The authors should, therefore, provide the readers with enough evidence as to why they consider these features to be resulting from sub-critical thermal fatigue, as they hypothesize, and how the stress measures that they calculated from the model support their conclusions.

We really appreciate the detailed comments from the Reviewer on this topic which have helped us to formulate a better understanding of how the manuscript can be improved. Our model does not allow for distinction between thermal fatigue- and shock-driven crack propagation, except in that it quantifies the magnitude of available stresses to drive said propagation. Based on the Reviewer's statements above (and in the previous round of revisions), they seem to agree that it

is reasonable to conclude that thermal fatigue is occurring on Bennu. From there, we had taken the logical step to conclude that, given the match to the model predictions for the location/direction of crack propagation, thermal fatigue is a possible driving mechanism to produce the exfoliation morphologies observed on Bennu. However, we formulate our response here based on the Reviewer's clarification that we did not adequately address the alternate scenario in which the stresses are too high for fatigue to occur, rather than too low. (The revisions to the actual manuscript are at the end of this section.)

Most planetary scientists with which we have discussed our work have had the opposite reaction, believing it to be unrealistic and inconsistent with general observations of asteroid surfaces for there to exist thermal stresses capable of causing shock (at least, for asteroid surfaces in near-Earth space or the Main Belt). A fairly common response to our work is along the lines of, "If thermal fracturing was really happening on Bennu, there wouldn't be any boulders left on the surface." For this reason, the tone of the wording in the manuscript was to convince the reader that stresses are indeed high enough to drive fatigue (e.g. Lines 203-204, "...even if these simulations overestimate stresses in real surface materials or a higher magnitude is needed to drive subcritical crack growth in vacuum"), even if they are not enough to drive shock.

In this light, one reason why we interpret fatigue as the more likely mechanism is that it occurs slowly and is therefore a more conservative interpretation of how (and how quickly) the asteroid surface is being modified. We feel the threshold of what constitutes *strong evidence* to show that a shock process is occurring is a higher bar to overcome. At this point, while the modeling results provide evidence that shock *could possibly* occur, we do not believe that evidence is strong enough to support the idea that shock is likely. For example, we do not have "before and after" images of rocks showing thermally disaggregated material or crack lengthening that clearly occurred over a short timescale.

The majority of terrestrial observations of boulder exfoliation attributed to thermal effects are driven by thermal fatigue, though there are limited observations of exfoliation and thermal spallation due to thermal shock from extreme (>620 K) temperature changes caused by forest fires, lightning, and (intentionally) via flame during mining operations. For example, Kendrick et al. (2016) observed that, resulting from a single forest fire, exfoliation layers of order centimeters thick were disaggregated across ~10-70% of the boulder surface area. This is consistent with the lower end of layer thicknesses that we observe on Bennu, however the timescale is improbable. If 1 cm of material was lost from a boulder each solar day on Bennu, then a 10 m boulder would only have a survival time of ~1000 Bennu days or ~0.5 Earth years. This is orders of magnitude smaller than 10^8 Earth years, our best lower-limit constraint on the amount of time Bennu has spent in near-Earth space (Walsh et al., 2019).

The Reviewer suggests that the features we observe could be the result of "intermittent and incremental shock fracture (repeated/cyclic dynamic crack growth and arrest)." Since the thermal stresses we model arise and subside in the boulders each day, an argument against intermittent fracture is a need for numerous thermal cycles to occur between shock events. On the other hand, they point out that, unlike in the case of failure by catastrophic "runaway" thermal shock, a crack could propagate a small amount each cycle and become arrested when moving into a lower-stress region of the boulder, resulting in incremental or recurrent crack

growth via shock. But, even in such a case where it took several cycles (even 1000, say) to lengthen a crack and fully disaggregate a 1 cm layer of material, the survival time of a 10 m boulder is still orders of magnitude smaller than 10^8 Earth years. Based on these crude calculations, it would require a net incremental crack growth rate of $<10^{-9}$ m per Bennu day in order for Bennu's largest (55 m) boulder to survive on the surface for 10^8 Earth years. This distance is smaller than the expected grain size of the rock, and far smaller than what would intuitively be expected for a thermal shock event (even a shock event which would not catastrophically destroy a rock).

Yet, Bennu's surface is completely dominated by boulders, larger and smaller, suggesting that erosion occurs substantially slower. Further, finer material (e.g., gravel, small rocks, dust), which would erode from the boulders as they break down, is not prevalent at Bennu's surface, as evidenced by the difficulty the team has had in finding a location where it is safe for the spacecraft to collect a sample. Shock experiments in the engineering literature support this idea. Wang et al. (2016) subjected sandstone to repeated thermal shocks and measured the sample degradation. The temperature change their samples were subjected to was similar in amplitude to that on Bennu (though applied more rapidly) and their samples were smaller. While not a perfect match to our scenario, the mechanical properties of sandstone are similar (same order of magnitude) to that of phyllosilicates, making the experiment a decent comparison. Wang et al. measured a ~50% decrease in the compressive strength and Young's modulus of sandstone subjected to repeated thermal shocks after just 40 cycles. This does not translate directly to an erosion rate, but it demonstrates how rapidly thermal shock processes degrade geologic materials.

In light of these arguments, we believe that a simpler, more conservative, and much more likely interpretation of the results is that the exfoliation is driven by fatigue, which is also consistent with the majority of terrestrial observations and with our understanding of the thermal environment of the asteroid's surface.

The Reviewer has suggested that we hedge our conclusion a little bit, using something along the lines of, "since the model shows that thermally-induced stresses build up in a way that aligns with the observed cracks, then a thermal origin is likely. However, the cracks may have been formed by rapid propagation at different times of the day (due to the high tensile stresses experienced at those instances), or by slow subcritical crack growth (fatigue)." We had tried to do this in the text, e.g. (Lines 193-195 of previous version) "It also suggests that thermal shock processes^{40,41}, which occur when an object's tensile strength is exceeded, could potentially play a role in weathering on Bennu's surface, though this is not supported by the observations (Figures 1-3) presented here." It is clear that this statement was insufficient, and so we have expanded the discussion in the text by replacing the line above with the following paragraph (new Lines 206-216):

"If Bennu's materials are sufficiently weak, it is plausible that thermal shock processes^{40,41}, which occur when an object's tensile strength is exceeded, could potentially drive the development of exfoliation fractures instead of fatigue. On the other hand, the rubble-pile asteroid is dominated by boulders with diameters ranging from tens of centimeters to tens of meters, with finer material such as gravel and dust covering only a small portion of its visible

surface (DellaGiustina and Emery et al., 2019; Walsh et al., 2019). This is not consistent with the rapid degradation and erosion rates associated with thermal shock events causing catastrophic (Kendrick et al., 2016) or incremental (via recurrent periods of crack growth and arrest) (Wang et al., 2016; Atkinson et al., 1984) rock failure, which would break down boulders quickly even over the shortest predicted timescale (100 Myr; Walsh et al., 2019) for Bennu's migration to near-Earth space. Thus, we interpret exfoliation via thermal fatigue to be the most likely origin of the observed features (Figure 1), which is consistent with terrestrial observations^{2,6,7}.

We have also tried to clarify how we present the upper limit stress magnitudes by changing Lines 186-188, "We take the magnitude of the local maximum stress as the upper limit of the exfoliating stress, which decreases to zero at the boulder surface" to read, "... which decreases by 20 to 50% at the exfoliation depth and reaches zero at the boulder surface."

Kendrick, K. J., Partin, C. A., & Graham, R. C. (2016). Granitic Boulder Erosion Caused by Chaparral Wildfire: Implications for Cosmogenic Radionuclide Dating of Bedrock Surfaces. *The Journal of Geology*, 124(4), 529–539. <http://doi.org/10.1086/686273>

Wang, P., Xu, J., Liu, S., & Wang, H. (2016). Dynamic mechanical properties and deterioration of red-sandstone subjected to repeated thermal shocks. *Engineering Geology*, 212, 1–9. <http://doi.org/10.1016/j.enggeo.2016.07.015>

Major Comment 2b: Atkinson (1984)

First, I commend the authors on adding a reference to Atkinson (1984), which gives a good background relevant to the authors' paper. I will note that Atkinson does not directly state that subcritical crack growth "requires stresses of ~20% of the material's tensile strength to occur", but rather "It is assumed that there exists a threshold K_{0} below which no significant crack extension can occur by stress corrosion. This is a function of the material's fracture properties and its environment. The value of K_{0} is [...] a small fraction (10-20%) of the critical stress intensity factor".

The authors' statement concerning the stress levels (and not stress intensities) and attributing the statement to Atkinson's work is, therefore, misleading, and the authors should consider rephrasing this sentence in their manuscript.

In fact, Atkinson devotes the largest part of the paper to discussing the different stress intensity factors and stress intensity thresholds. The very first equation in Atkinson's paper emphasizes that the near-field stresses around the crack are enhanced, and they make little mention of the material's "tensile strength", but rather focus on the material's "fracture toughness".

I recommend the authors refer to other supporting literature as well, such as Emmerich et al. (2007), where it is demonstrated that the relationship between fracture toughness and tensile strength is not as straight-forward as the authors seem to imply.

Emmerich, F. G. (2007). Tensile strength and fracture toughness of brittle materials. *Journal of Applied Physics*, 102(7), 073504.

Furthermore, based on the authors' response, the authors seem to think the stress intensity factor can be directly estimated from the "maximum principal stress multiplied by a geometrical factor". I assume they are basing this argument on the far-field analysis that is, for example, the subject of equation #2 in Atkinson et al. (1984)'s paper. However, I will bring to the authors' attention that this statement is true for a remotely-applied stress field, where the stress state everywhere away from the crack tip is uniform. In the case of thermal stress, there is a local stress state that is varying in both space (stress gradients) and time along both the crack tip and the crack faces, which gives rise to a complex mixed-mode loading, and so the stress intensity factor is not easily estimated from the peak stresses alone.

This, once more, highlights why the reliance on the maximum tensile stress as the only metric is not as straight-forward when discussed in the context of thermal fatigue in heterogeneous and flaw-ridden geologic materials. I recommend the authors refer to the relevant work in literature, especially the works of Nemat-Nasser, to cite a few:

Nemat-Nasser, S., and Oranratnachai, A. (March 1, 1979). "Minimum Spacing of Thermally Induced Cracks in Brittle Solids." *ASME. J. Energy Resour. Technol.* March 1979; 101(1): 34–40. <https://doi.org/10.1115/1.3446859>

Geyer, J. F., & Nemat-Nasser, S. (1982). Experimental investigation of thermally induced interacting cracks in brittle solids. *International Journal of Solids and Structures*, 18(4), 349-356.

These are excellent and thoughtful references, and we thank the Reviewer for their detailed discussion on this. We are aware of the fact that, in many scenarios, the stress intensity factor does not have a straightforward relationship to macroscopic stress, and the Reviewer's point about the over-simplicity and wording of the statement regarding the "~20% of the material's tensile strength" in the text is well taken. We do not wish to mislead our readers. We explain our thought process and reasoning for our approach below, detailing the manuscript revisions at the end.

Given the scope of this study, we feel it is a reasonable first order approximation to consider the scenario for a mode 1 (tensile) crack propagation in an infinite medium where stress is related to stress intensity using a far-field approach (e.g., Atkinson's equation 2). Thermal stress in these boulders does vary spatially and temporally, making reality somewhat more complex, but the aim of this study is to substantiate the claim that thermal fatigue can (and does) occur on rocks on the asteroid surface, and to this end we need only to establish the stresses are high enough to drive it. We explicitly chose not to use a more detailed crack propagation model because there are too many uncertainties about the material properties and microstructure, many of which

(hopefully!) will be illuminated when the sample OSIRIS-Rex will collect from the asteroid surface is returned to Earth.

We make the approximation that, in a large boulder, the stress field local to a given microcrack will be quasistatic for some period of the day during which the crack is experiencing the highest stress it will experience throughout the day. The stress around the microcrack is, of course, changing, but we endeavor to make a determination whether or not, at *some* time during the day, the stress is high enough to cause it to propagate. For this reason, we use the highest maximum principal stress experienced by the crack as a proxy for the most amount of available energy able to go towards propagating it. We are not calculating the rate of propagation, and so we are not concerned with magnitude of the stress except that it can be reasonably assumed to be above the critical threshold required for fatigue.

For some materials, real heterogeneities of the material microstructure may lead mixed-mode to fracturing, however we have already made it clear why we are not considering complexities at this level of detail in this study. Our boulders are idealized in shape and microstructure homogeneity to understand, at the simplest level, how they are expected to respond to thermal forcing from the Sun. As a first order approximation and, based on the orientation of the macroscopic stress fields, it is clear that the thermally induced stress fields will tend to drive mode I (tensile) cracks that propagate parallel to the boulder surface.

As discussed in detail by Emmerich (2007) the fracture toughness cannot always be related to the tensile strength in a straightforward manner, and in this context using the maximum principal stress as our metric is not perfect. However, again considering that our aim here is simply to establish that fatigue can occur, we can use maximum principal stress to make a first order approximation as to whether the criterion is met. Since the boulder is large compared to the size of the microcrack (crack-parallel dimension is \gg than the crack width), it is reasonable to assume that the stress intensity factor follows the classic case for Griffith's criterion of a penny shaped crack in an infinite domain. In this simple scenario (e.g., Atkinson's equation 2), the stress intensity factor is proportional to the macroscopic stress, multiplied by factor of π and the crack length. As Emmerich (2007) points out, in this simple case the critical stress intensity factor at which (critical) fracture occurs (that is, the fracture toughness) is then equal to the material tensile strength times the factor of π and the crack length. Atkinson notes that the critical threshold required for fatigue to occur is "a small fraction (10-20%) of the critical stress intensity factor," and therefore a value of 20% of the critical stress intensity translates to a value of 20% of the macroscopic material strength.

We have replaced the sentence in the manuscript referred to by the Reviewer above with the following (Lines 196-204):

"Crack propagation models express stress fields in terms of the stress intensity factor, which relates the macroscopic stress field to the stress around the crack tip. The threshold to drive sub-critical crack growth in terrestrial environments typically requires a stress intensity factor that is \sim 10 to 20% of the material's fracture toughness (Atkinson, 1984). To first order, surface-parallel microcracks can be approximated as cracks in an infinite medium, where the material's tensile strength is linearly proportional to its fracture toughness (Emmerich, 2007). Our results show we

have sufficient stress to overcome a threshold of 20% of the tensile strength, indicating that fatigue is possible on Bennu...”

Major Comment 2c [Comment already addressed to Reviewer’s satisfaction]

The major concern voiced in my first review was that I did not agree with the authors that their work “confirms the oft-positied but previously unverified hypothesis that thermal fatigue is an active process on airless bodies”, especially since the model itself does not provide much evidence to “confirm” and “verify” the hypothesis. I commend them for softening the statement to “substantiates the oft-positied hypothesis that thermal fatigue is an active process on airless body surfaces”.

Two different reviewers voiced similar opinions on this statement, and in re-reading we agreed that a softer verb was more appropriate. We are glad the Reviewer is satisfied with this change.

Major Comment 2d: Scale-dependent Strength

Second, what is the frequency distribution of the strengths for rocks of similar sizes to the ones they are studying in the models? That is, what is the probability of finding a meter-sized boulder with a tensile strength of 5 MPa (their upper range)? Also, why should we not expect that tensile strength of boulders on Bennu be comparable to the tensile strength estimates of boulders on Ryugu (0.2 MPa)?

One of the reasons why brittle geologic materials have such a large distribution of tensile strengths is due to their sub-scale heterogeneities (micro-scale flaws and cracks). The larger boulders (their meter-sized simulations) would, therefore, be more likely to be on the lower-end of the strength scale as they would have a larger distribution of flaws. See for example the strength scaling work done in literature for asteroid impact simulations:

Housen, K. R., & Holsapple, K. A. (1999). Scale effects in strength-dominated collisions of rocky asteroids. *Icarus*, 142(1), 21-33.

Holsapple, K., Giblin, I., Housen, K., Nakamura, A., & Ryan, E. (2002). Asteroid impacts: Laboratory experiments and scaling laws. *Asteroids III*, 1, 443-462.

Another paper that may be of interest to the authors is Cotto-Figueroa et al. (2016) Cotto-Figueroa, D., Asphaug, E., Garvie, L. A., Rai, A., Johnston, J., Borkowski, L. & Morris, M. A. (2016). Scale-dependent measurements of meteorite strength: Implications for asteroid fragmentation. *Icarus*, 277, 73-77.

Realizing that this paper deals with compressive strength (although a similar conclusion can be extracted for tensile strength), the authors find that “the common use of terrestrial analog materials to predict scale-dependent strength properties significantly overestimates the strength of meter-sized asteroidal materials” and “boulders of similar

composition on asteroids will have compressive strengths significantly less than typical terrestrial rocks”.

We appreciate the helpful references provided by the Reviewer on the topic of the scale dependency of material strength, one of which was new to us. The implications for scale-dependent strength is discussed by Molaro et al. (2015) and Molaro et al. (2017) with respect to idealized simulations of thermal stresses in airless body materials.

We do not know the “probability of finding a meter-sized boulder with a tensile strength of 5 MPa” on Bennu, as we have very few constraints on the thermophysical and mechanical properties of Bennu’s surface materials. Observations indicate, generally, that their composition is dominated by hydrated phyllosilicates, and their varied morphologies across the surface point to a range of properties such as porosity and thermal conductivity. There is still much work to be done before we can understand the boulder population in more detail than we have described here. At this stage, our best understanding of material strengths comes from comparison to analog materials and measurements of carbonaceous chondrite meteorites. A more complex understanding will emerge over the coming months and years as the team analyzes the data we are collecting at the asteroid. It is exactly for this reason that we take the simpler approach of quantifying the stress magnitudes with an elastic model rather than a more detailed crack model, and rely on a maximum principal stress criterion to establish, to first order, whether or not fatigue may occur.

We have not performed a comparative analysis between what is currently known about Bennu versus Ryugu’s boulder compositions and morphologies. We do not have access to data from the Hayabusa 2 team, but to our knowledge, exfoliation features have not been observed on Ryugu and there are some differences in boulder composition. I am sure that opportunities for such a comparison will be facilitated as more data is eventually released to the public over time.

Major Comment 3: Static Fatigue and Effect of Pressure

The authors state that “laboratory studies show that thermally driven crack propagation is harder to achieve in anhydrous and vacuum environments than in ambient atmosphere” (line 62), but the experimental procedure in the referenced paper does not appear to “thermally-drive” cracks, but rather performed compression and tensile tests, in addition to a “static fatigue” test, where according to the paper “the specimens were loaded to a stress calculated to be roughly 90% of their fracture strength and held at this stress until fracture occurred.” (that is, the stress magnitude was not varied). I suggest the authors rephrase this sentence to properly reflect what has been tested in Krokosky’s paper, as it does not seem that Krokosky examined thermally-driven crack propagation.

I also recommend the authors refer to Atkinson et al. (1984)’s discussion on the effect of pressure on the fatigue lifetime of rocks, as this discussion is relevant to how vacuum (or the lack of pressure) may influence the fatigue thresholds.

We thank the Reviewer for these notes. It is difficult to find studies of fatigue crack growth for geologic materials and under vacuum conditions and for thermal (cyclic) fatigue. While the Reviewer's point is noted here, we also do not wish to neglect references we feel will be useful to readers. Eppes and Keanini (2017) find that, under certain conditions, the empirical parameters quantified by laboratory studies for static and cyclic sub-critical crack growth may be interchanged. We have elected to leave the reference to Krokosoky and Husak (1968) (and the related text), but have added a reference to both Atkinson (1984) and Eppes and Keanini (2017) to round out the scope of literature reflected by the citation.

Eppes, M. C., & Keanini, R. (2017). Mechanical weathering and rock erosion by climate-dependent subcritical cracking. *Reviews of Geophysics*, 55(2), 470–508. <http://doi.org/10.1002/2017RG000557>

Minor Comments:

[deleted text copied from the author's previous response]

I will once again quote Atkinson et al. (1984)'s paper "At stress intensity factors above the stress corrosion crack growth limit, significant crack growth will be suppressed if the stress is high enough for diffusion creep to make a significant contribution to deformation rate and thereby blunt macrocrack tips. This has been observed in ceramics."

I recommend that, in their analysis, the authors acknowledge, at the very least, that such crack-tip blunting mechanisms may occur. If such a blunting were to occur, then the cracks' fatigue growth may even be arrested. One can then make use of such a possibility to hypothesize that the features the authors are examining may have not been formed by thermal fatigue, but rather by a rapid crack growth happening at specific times of the day where the peak stresses are reached.

The Reviewer's note on crack blunting is appreciated here, in the context of the possibility of shock-driven cracks becoming arrested. We have elected not to mention crack blunting specifically because we ultimately still find thermal shock to be an unlikely scenario. However, we have included a reference to Atkinson et al. in the paragraph where we mention crack arresting.

Formatting Comments:

There is a typo in line 73. The reference #2 needs to be a superscript.

We were unable to find a typo in Line 73 (or a nearby line) and assume there is another line number mismatch. We will look out for this typo during proofing should the article be accepted.

I suggest the "ref." on line 367 to be changed to DellaGiustina et al., to be consistent with the referencing style used elsewhere (line 169, for example).

This has been corrected.

Reviewers' Comments:

Reviewer #3:

Remarks to the Author:

Reviewer #3:

I have reviewed the authors' responses and revisions made to the manuscript, and I think the authors have responded adequately to all my comments.

I therefore think that the manuscript is suitable for publication, and I commend the authors on their valuable work.

General Comments

The authors indicated that the line numbers mentioned in my review do not match with the manuscript. In my copy, the line numbering starts from the title, and the first sentence in the manuscript after the abstract ("thermally induced breakdown or thermal stress weathering ..."), is numbered as line 30. What the authors say is line 190 in their copy ("...estimates of the tensile strength of boulders on (162173) Ryugu...") appears as line 219 on my copy. Therefore, it seems that there is an offset of around 30 lines in our numbering schemes, which may correspond to where the line numbering begins on each document. This is also consistent with the offsets indicated later on, where Line 213 on my copy is line 185 on theirs, and line 62 on my copy is line 35 on theirs.

I suggest that the authors re-read my comments with this in mind to ensure that all lines are correctly identified.

Concerning the authors' clarification on what they interpreted as my intention in the first review, I would like to remind the authors that in the original document, they claimed that "this work confirms the oft-positied but previously unverified hypothesis that thermal fatigue is an active process on airless body surfaces."

It is therefore only natural to require a more stringent methodology if the authors want to hold the claim that their work (observations coupled with modeling) confirms and verifies beyond reasonable doubt the thermal fatigue hypothesis. The model in its current state cannot satisfactorily answer the many questions required to really "confirm" and "verify" the hypothesis. The authors claimed that the "if the Reviewer agrees that the features cannot be explained by impacts, then by process of elimination, thermal fatigue is the most feasible mechanism"; however, my response demonstrated an alternative way one can interpret the authors' same results by claiming it is thermal shock, and not fatigue, that drives the crack growth. The stress levels, especially given how comparable they are to the supposed material's strength, could be used to claim thermal shock is the "most feasible mechanism, by process of elimination". Thus, I maintain that the authors' model as it is right now does not provide the sufficient evidence to "confirm" the "previously unverified hypothesis" of thermal fatigue.

The authors seem to have reached this same conclusion in their responses, as well, as they attempted to provide temporal estimates to dismiss the possibility of a thermal shock-driven crack growth.

Since the authors' initial claim has been softened in the revised manuscript ("substantiates" the hypothesis), then the current methodology along with the additional explanation regarding the model is acceptable and a good addition to the manuscript, as the authors agree.

REVIEW RESPONSE

The authors indicated that the line numbers mentioned in my review do not match with the manuscript. In my copy, the line numbering starts from the title, and the first sentence in the manuscript after the abstract (“thermally induced breakdown or thermal stress weathering ...”), is numbered as line 30. What the authors say is line 190 in their copy (“...estimates of the tensile strength of boulders on (162173) Ryugu...”) appears as line 219 on my copy. Therefore, it seems that there is an offset of around 30 lines in our numbering schemes, which may correspond to where the line numbering begins on each document. This is also consistent with the offsets indicated later on, where Line 213 on my copy is line 185 on theirs, and line 62 on my copy is line 35 on theirs. I suggest that the authors re-read my comments with this in mind to ensure that all lines are correctly identified.

We have reviewed the previous review to identify where line numbers were provided for reference. The following comments included line numbers which we had been confused about:

[Part of Major Comment 1] However, the authors must also acknowledge that their model constitutes a ‘guide’ for their ‘best guess’ on how thermally-induced cracks ‘may’ grow (eg. line 190).

[A part of Major Comment 2a] The authors state that “Subcritical crack growth on Earth only requires stresses of ~20% of the material’s tensile strength to occur”, and acknowledge that “once a macroscale feature has developed, stresses become enhanced at the crack tip” (line 213). The authors also admit that “the upper limits” of their maximum tensile stresses are “comparable to the tensile strengths of terrestrial phyllosilicate rocks” (line 217) and “exceeds the estimate of the tensile strength of boulders on Ryugu”.

[A part of Major Comment 3] The authors state that “laboratory studies show that thermally driven crack propagation is harder to achieve in anhydrous and vacuum environments than in ambient atmosphere” (line 62)...

All of these comments were successfully addressed by our previous changes, regardless of confusion over line numbers.

There is a typo in line 73. The reference #2 needs to be a superscript.

We did not find a typo, but assume it was already found during general editing. We have now corrected the superscript error.

I suggest the “ref.” on line 367 to be changed to DellaGiustina et al., to be consistent with the referencing style used elsewhere (line 169, for example).

This error was found by searching the document for the word “ref.” and has already been corrected.